# Structural insights into two distinct binding modules for Lys63-linked polyubiquitin chains in RNF168

Tomio S. Takahashi[1,2], Yoshihiro Hirade [3], Aya Toma[1,2,4], Yusuke Sato[1,2,4], Atsushi Yamagata[1,2,4], Sakurako Goto-Ito[1,2], Akiko Tomita[3], Shinichiro Nakada[3,5] & Shuya Fukai [1,2,4]

The E3 ubiquitin (Ub) ligase RNF168 plays a critical role in the initiation of the DNA damage response to double-strand breaks (DSBs). The recruitment of RNF168 by ubiquitylated targets involves two distinct regions, Ub-dependent DSB recruitment module (UDM) 1 and UDM2. Here we report the crystal structures of the complex between UDM1 and Lys63-linked diUb ($K63-Ub_2$) and that between the C-terminally truncated UDM2 (UDM2ΔC) and $K63-Ub_2$. In both structures, UDM1 and UDM2ΔC fold as a single α-helix. Their simultaneous bindings to the distal and proximal Ub moieties provide specificity for Lys63-linked Ub chains. Structural and biochemical analyses of UDM1 elucidate an Ub-binding mechanism between UDM1 and polyubiquitylated targets. Mutations of Ub-interacting residues in UDM2 prevent the accumulation of RNF168 to DSB sites in U2OS cells, whereas those in UDM1 have little effect, suggesting that the interaction of UDM2 with ubiquitylated and polyubiquitylated targets mainly contributes to the RNF168 recruitment.

[1] Institute of Molecular and Cellular Biosciences, The University of Tokyo, Tokyo 113-0032, Japan. [2] Synchrotron Radiation Research Organization, The University of Tokyo, Tokyo 113-0032, Japan. [3] Department of Bioregulation and Cellular Response, Graduate School of Medicine, Osaka University, Osaka 565-0871, Japan. [4] Department of Computational Biology and Medical Sciences, Graduate School of Frontier Sciences, The University of Tokyo, Chiba 277-8501, Japan. [5] Institute for Advanced Co-Creation Studies, Osaka University, Osaka 565-0871, Japan. Correspondence and requests for materials should be addressed to S.N.(email: snakada@bcr.med.osaka-u.ac.jp) or to S.F. (email: fukai@iam.u-tokyo.ac.jp)

Ubiquitin (Ub) is a highly conserved 76-residue protein, which can be covalently attached to substrate proteins to regulate various cellular events[1]. The side-chain amino groups of seven lysine residues (Lys6, Lys11, Lys27, Lys29, Lys33, Lys48, and Lys63) or the terminal amino group of Met1 in one Ub can be bonded to the terminal carboxyl group of another Ub, producing eight structurally distinct types of Ub chains[2]. Differences in the length, linkage and branching of Ub chains increase the complexity of the Ub signaling system. MonoUb or Ub chains on substrate proteins are recognized by proteins called Ub receptors, which contain one or more Ub-binding domains (UBD). There are at least 21 types of UBDs[3, 4]. In some cases, the co-operative binding between multiple UBDs and Ub moieties confers chain type specificities on Ub receptors through an avidity-based mechanism[5].

In DNA damage response, a cascade of mono- and poly-ubiquitylation processes activates different pathways of DNA damage signaling and DNA repair[6]. DNA double-strand breaks (DSBs) are one particularly toxic type of DNA lesions. The process of DSB repair requires recruitment of the E3 Ub ligase RNF8 by ATM-phosphorylated MDC1[7–9]. RNF8 together with UBC13 promotes the formation of Lys63-linked Ub chains (hereafter referred to as K63 chains) on chromatin-associated proteins including linker histone H1[8, 10–13]. In the current model, the E3 Ub ligase RNF168 recognizes these polyubiquitylated proteins and then ubiquitylates histone H2A, which serves as a recruitment signal for 53BP1[14–19]. RNF168 can also bind the products of its own Ub ligase activity, facilitating its accumulation at DSB sites[20]. The third RING finger protein RNF169, a paralog of RNF168, is also recruited by RNF168-ubiquitylated targets, although it is controversial whether the Ub ligase activity of RNF169 is physiologically relevant to DSB repair[20–22].

Functional importance of RNF168 in the process of DSB repair has been demonstrated by its mutations associated with the RIDDLE (radiosensitivity, immunodeficiency, dysmorphic features, and learning difficulties) syndrome[16]. Recognition of ubiquitylated products by RNF168 relies on three distinct UBDs: MIU (motif interacting with Ub) 1, MIU2 and UMI (Ub-interacting motif [UIM]- and MIU-related UBD) (Fig. 1). These domains are essential for recruitment of RNF168[14–16, 23]. UMI and MIU1 are included in a module called UDM (Ub-dependent DSB recruitment module) 1, which is located in the N-terminal part of RNF168. On the other hand, MIU2 is included in a module called UDM2, which is located in the C-terminal part of RNF168 (Fig. 1). Currently, it is postulated that UDM1 and UDM2 interact with different ubiquitylated targets: UDM1 is associated with RNF8-dependent polyubiquitylated targets, whereas UDM2 recognizes RNF168-dependent mono-ubiquitylated targets[13, 20]. The functional difference between UDM1 and UDM2 may be linked to the conserved motifs named LR motifs (LRMs)[20].

LRMs were initially identified as important elements for the recruitment of RNF168 and RNF169[20]. Both proteins can bind

RNF168-dependent ubiquitylated targets, such as H2A. This interaction is mediated by a common motif LRM2, which is located in the C-terminal part of UDM2[20]. This conserved motif is required for the recruitment of RNF168 and RNF169 to ubiquitylated H2A and seems to bind directly to H2A without affecting Ub-binding properties of UDM2[20, 24]. LRM1 was identified as a binding element for proteins ubiquitylated by RNF8[20]. Further study showed that the acidic patch of LRM1 in UDM1 could interact with linker histone H1[13]. Surprisingly, RNF169, which is not recruited by RNF8-mediated ubiquitylation and should not interact with H1, still possesses a highly conserved LRM1 motif, which raises questions concerning the function of this motif. In addition, an RNF168 region containing both LRM1 and UMI was found to display stronger affinity for K63 chains than UMI alone, suggesting that LRM1 might have an additional role in the interaction with Ub[23].

Although, functional studies on RNF168 have demonstrated that its recognition of K63 chains is a critically important process in DNA damage response, the underlying structural mechanism remains elusive. Here we present the crystal structures of UDM1 and the C-terminally truncated UDM2 (UDM2ΔC) of RNF168 in complex with Lys63-linked diUb (K63-Ub$_2$) at 1.78 and 1.80 Å resolutions, respectively. Together with structure-based muta-genesis and binding analyses using surface-plasmon resonance (SPR) spectroscopy, we elucidate how UDM1 and UDM2 of RNF168 specifically interact with K63 chains. LRM1 and an additional motif in UDM2 were defined as essential binding elements for the distal Ub (Ub$^{dist}$) linked to Lys63 of the proximal Ub (Ub$^{prox}$). Further analyses of the recruitment of RNF168 mutants in gamma-irradiated cells dissect the functional roles of UDM1 and UDM2 in the context of the Ub signaling for DNA damage response.

## Results

**Characterization of RNF168 UDM1**. Previous studies have shown that RNF168 UDM1 comprises UMI (residues 141–156)[23] and MIU1 (residues 171–188)[14–16] as Ub-binding motifs. It has also been assumed that these two adjoining Ub-binding motifs could confer the specificity for K63 chains to UDM1, probably by analogy to the tandem UIM motifs of RAP80 and Epsin1[5, 25–27]. However, our pull-down analysis suggested that UMI–MIU1 (residues 134–188) bound similarly to Met1-, Lys6-, Lys11-, Lys29-, Lys33-, and Lys48-linked diUb species (M1-, K6-, K11-, K29-, K33-, and K48-Ub$_2$, respectively) and K63-Ub$_2$ (Supplementary Fig. 1a). In contrast, the N-terminal part of UDM1 (residues 110–166; LRM1–UMI) bound preferentially to K63-Ub$_2$ (Supplementary Fig. 1a). This observation is consistent with a previous finding that residues 56–166 of RNF168, which contain both LRM1 and UMI but not MIU1, bind specifically to K63 chains[23]. Another study showed that the full-length RNF168 was able to bind efficiently to Lys27-linked diUb (K27-Ub$_2$)[28]. We therefore tested the binding of RNF168 to K27-Ub$_2$ by pull-down analysis. The binding of LRM–UMI to K27-Ub$_2$ was much weaker than that to K63-Ub$_2$, whereas UMI–MIU1 bound similarly to M1-, K27- and K63-Ub$_2$ species (Supplementary Fig. 1b).

We then examined the binding of the GST-tagged LRM1, UMI, LRM1–UMI or MIU1 to K63-Ub$_2$ by SPR spectroscopy (Table 1, Supplementary Fig. 1c). Their binding to K48- or M1-Ub$_2$ was also examined as non-cognate controls. LRM1 exhibited no binding to K63-, K48-, or M1-Ub$_2$, whereas UMI bound K63-, M1-, and K48-Ub$_2$ with $K_d$ values of 283, 306, and 653 μM, respectively. These results are consistent with the previous finding that UMI alone binds long K63 chains more efficiently than long Lys48-linked Ub chains (hereafter referred to as K48 chains)[23]. On the other hand, LRM1–UMI exhibited a substantially stronger

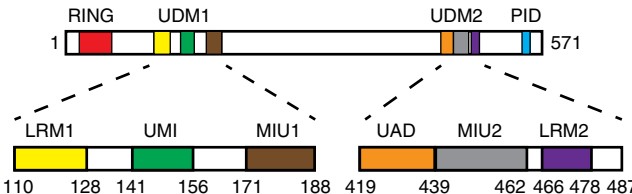

**Fig. 1** Functional motifs in human RNF168. RING, LRM1, UMI, MIU1, UAD, MIU2, LRM2, and PALB2-interacting domain (PID) are shown as red, yellow, green, brown, orange, gray, purple, and light blue boxes, respectively

**Table 1 Binding affinity of UMI, LRM1, LRM1–UMI, MIU2, UAD, or UDM2ΔC for K63-, K48-, or M1-Ub$_2$**

|  |  | $K_d$ (μM) | | |
|---|---|---|---|---|
|  |  | K63-Ub$_2$ | M1-Ub$_2$ | K48-Ub$_2$ |
| UDM1 | UMI | 283 ± 12 | 306 ± 9 | 653 ± 27 |
|  | LRM1 | ND | ND | ND |
|  | LRM1-UMI | 45 ± 5 | 231 ± 43 | 435 ± 37 |
|  | LRM1-UMI (untagged)[a] | 42 ± 1 | 200 ± 8 | 316 ± 9 |
|  | LRM1 (L116A)-UMI | 136 ± 1 | 206 ± 11 | 389 ± 39 |
|  | LRM1 (Y120A)-UMI | 168 ± 22 | 236 ± 23 | 405 ± 37 |
|  | LRM1-UMI (S142A) | 390 ± 74 | 965 ± 415 | ND |
|  | LRM1-UMI (L149A) | 486 ± 106 | 1150 ± 619 | ND |
|  | MIU1 | 188 ± 3 | 233 ± 13 | 568 ± 74 |
|  | MIU1 (D175A) | ND | ND | ND |
| UDM2 | MIU2 | 147 ± 3 | 235 ± 16 | ND |
|  | UAD | ND | ND | ND |
|  | UDM2 | 34 ± 6 | 238 ± 16 | 298 ± 12 |
|  | UDM2ΔC | 25 ± 4 | 221 ± 9 | 301 ± 8 |
|  | UDM2ΔC (untagged)[a] | 18 ± 0.05 | 145 ± 1 | 253 ± 3 |
|  | UDM2ΔC (E433R) | 107 ± 2 | 214 ± 18 | 474 ± 65 |
|  | UDM2ΔC (L436T) | 77 ± 6 | 160 ± 2 | 598 ± 52 |
|  | UDM2ΔC (R439A) | 196 ± 13 | 495 ± 46 | ND |
|  | UDM2ΔC (D446A) | ND | ND | ND |

ND not detectable
Data are presented as mean ± standard deviation; $n = 3$ independent experiments
[a] $K_d$ was determined using untagged protein

affinity to K63-Ub$_2$ ($K_d = 45$ μM) than to M1- or K48-Ub$_2$ ($K_d = 231$ μM or 435 μM, respectively). Note that we confirmed that the presence of the GST tag did not confer Lys63-linkage specificity on LRM1–UMI by analyzing the binding of the untagged LRM1–UMI to K63-, M1- or K48-Ub$_2$ ($K_d = 42$, 200, or 316 μM, respectively), because a previous study reported that a dimeric property of the GST tag affected the linkage specificity of some UBA domains[29]. These results suggest that LRM1 serves as the auxiliary element for UMI to specifically recognize K63 chains. In the C-terminal part of UDM1, MIU1 alone (residues 161–194) bound to K63-, M1-, or K48-Ub$_2$ with $K_d$ of 188, 233, or 568 μM, respectively, indicating that MIU1 has no linkage specificity for Ub chains, consistently with a previous study[30].

**Structure of RNF168 UDM1 in complex with K63-Ub$_2$.** To confirm the specific interaction between RNF168 LRM1–UMI and K63 chains and understand its underlying structural mechanism, we determined the crystal structures of RNF168 UDM1 in complex with K63-Ub$_2$ in three distinct crystal forms (Table 2, Fig. 2a, Supplementary Fig. 2a,b). The structures were determined by molecular replacement, using the Rabex-5 MIU–Ub complex (PDB 2FID) as the search model[31]. One of the three forms is totally different from the other two forms: two UDM1 molecules and two K63-Ub$_2$ molecules are assembled into a tetrameric complex, where Ub$^{dist}$ and Ub$^{prox}$ of K63-Ub$_2$ interact with two distinct UDM1 molecules (1130 Å$^2$ buried surface area averaged over the two tetrameric complexes in the asymmetric unit) (Supplementary Fig. 2a). This tetrameric assembly (theoretical molar mass of 54 kDa) is inconsistent with the molar mass of the RNF168 UDM1–K63-Ub$_2$ complex (27 kDa) determined experimentally by size-exclusion chromatography coupled with multi-angle light scattering (SEC-MALS) (Supplementary Fig. 2c). Therefore, we concluded that this crystal form is an artifact generated during crystallization. On the other hand, the other two crystal forms (forms I and II at 1.78 and 2.25 Å resolutions, respectively) contain two stoichiometric UDM1–K63-Ub$_2$ complexes in the asymmetric unit (Supplementary Figs. 2b and 3). No large conformational difference in

UDM1 or K63-Ub$_2$ was observed between the two crystal forms or between the two molecules in the asymmetric unit (Supplementary Fig. 3a). LRM1, UMI, and MIU1 fold into a single-continuous α-helix, whereas K63-Ub$_2$ adopts an extended conformation (Fig. 2a, Supplementary Fig. 2b). LRM1–UMI binds K63-Ub$_2$, where Ub$^{dist}$ and Ub$^{prox}$ interact with LRM1 and UMI, respectively (1250 Å$^2$ and 1090 Å$^2$ buried surface areas in forms I and II, respectively, averaged over the two complexes in the asymmetric unit) (Fig. 2a, Supplementary Fig. 2b). On the other hand, MIU1 binds Ub$^{dist}$ of the adjacent K63-Ub$_2$ in the crystal (Fig. 2a, Supplementary Fig. 2b). Electron density corresponding to the linkage between Gly76 of Ub$^{dist}$ and Lys63 of Ub$^{prox}$ in K63-Ub$_2$ is discontinuous, suggesting its structural flexibility. The interdomain region between LRM1 and UMI appears unable to interact with K63-Ub$_2$. All these findings support the idea that LRM1–UMI serves as a functional unit for specific recognition of K63 chains within UDM1.

**Recognition of K63-Ub$_2$ by LRM1–UMI of RNF168 UDM1.** Leu116 and Tyr120 of RNF168 LRM1 interact with the Ile36-centered hydrophobic patch of Ub$^{dist}$ in K63-Ub$_2$ (Fig. 2b, left). The side chain of RNF168 Leu116 is inserted into a hydrophobic pocket formed by Ile13, Ile36, and Leu69 and the aliphatic portions of Thr7 and Glu34 in Ub$^{dist}$. Tyr120 of RNF168 hydrophobically interacts with Ile36, Leu71, and Leu73 of Ub$^{dist}$ (Fig. 2b, left). In addition, Glu119 of RNF168 hydrogen bonds with Thr9 of Ub$^{dist}$. The main-chain amide groups of RNF168 Ser111 and Arg117 hydrogen bond with the main-chain carbonyl group of Thr12 and the side chain of Glu34 in Ub$^{dist}$, respectively (Fig. 2b, right). Note that the Ser111-mediated interaction with Ub$^{dist}$ is missing in the form II crystal, since it contains only residues 113–188 of RNF168 (Supplementary Fig. 3a). The Ala replacement of the conserved Leu116 or Tyr120 of RNF168 decreased the affinity of LRM1–UMI for K63-Ub$_2$ to 33 or 27% of wild-type affinity, respectively, with little effect on the affinity for M1- or K48-Ub$_2$ (Table 1, Supplementary Fig. 1c).

RNF168 UMI interacts with Ub$^{prox}$ of K63-Ub$_2$ (Fig. 2c). Tyr145, Ile146, Leu149, and Leu150 of RNF168 form a hydrophobic surface to interact with the Ile44-centered hydrophobic patch of Ub$^{prox}$ (Fig. 2c, left). This hydrophobic interaction is further stabilized by three hydrogen bonds: Tyr145, Glu153, and Ser142 of RNF168 hydrogen bond with Gln49, Arg42, and the Gly47 main-chain amide group of Ub$^{prox}$, respectively (Fig. 2c, right). The Ub$^{prox}$-interacting residues of RNF168 (i.e., Ser142, Tyr145, Ile146, Leu149, Leu150, and Glu153) are completely conserved among representative vertebrates (Fig. 2d). The Ala replacement of Ser142 or Leu149 decreased the affinity of LRM1–UMI for K63-Ub$_2$ to 12 or 9% of wild-type affinity, respectively (Table 1, Supplementary Fig. 1c). The Ala replacement of Ser142 or Leu149 also decreased the affinity for M1-Ub$_2$ to 24 or 20% of wild-type affinity, respectively, and eliminated the binding to K48-Ub$_2$ (Table 1, Supplementary Fig. 1c).

Taken together, the K63-linkage specificity of RNF168 LRM1–UMI depends on the interaction with Ub$^{dist}$ that is linked to Lys63 of Ub$^{prox}$. This feature is in striking contrast to other linkage-specific Ub receptors or DUBs, where the interaction with Ub$^{prox}$ defines their linkage specificities[32].

**Characterization of RNF168 UDM2.** A previous study has shown that RNF168 UDM2 (residues 419–487) preferentially binds K63-Ub$_2$, although its binding seems weaker than that between UDM1 and K63-Ub$_2$[13]. MIU2 (residues 436–462) is the only Ub-binding motif that has been identified in UDM2. Our SPR analysis showed no obvious difference between the affinity of

**Table 2 Data collection and refinement statistics**

| | RNF168 UDM1–K63-Ub₂ (form I) | RNF168 UDM1–K63-Ub₂ (form II) | RNF168 UDM1–K63-Ub₂ (tetrameric form) | RNF168 UDM2ΔC–K63-Ub₂ |
|---|---|---|---|---|
| *Data collection* | | | | |
| Beamline | SPring-8 BL41XU | SPring-8 BL41XU | SPring-8 BL41XU | SPring-8 BL41XU |
| Wavelength (Å) | 1.0000 | 1.0000 | 1.0000 | 1.0000 |
| Space group | $P1$ | $P1$ | $P2_1$ | $P2_12_12$ |
| Cell dimensions | | | | |
| $a, b, c$ (Å) | 35.3, 66.3, 74.2 | 45.4, 50.0, 64.4 | 85.3, 64.1, 117.5 | 59.4, 78.5, 31.0 |
| $\alpha, \beta, \gamma$ (°) | 76.5, 79.3, 80.5 | 73.5, 69.7, 73.8 | 90.0, 109.6, 90.0 | 90.0, 90.0, 90.0 |
| Resolution (Å) | 50.0–1.78 (1.81–1.78) | 50.0–2.25 (2.29–2.25) | 50.0–2.50 (2.54–2.50) | 50.0–1.80 (1.83–1.80) |
| Completeness (%) | 92.9 (92.0) | 90.2 (91.5) | 97.2 (97.4) | 99.3 (99.2) |
| Redundancy | 6.3 (5.0) | 3.1 (3.0) | 5.1 (4.4) | 7.7 (6.3) |
| $I/\sigma I$ | 32.7 (1.4) | 20.2 (1.8) | 6.9 (1.3) | 20.7 (2.7) |
| $R_{sym}$ | 0.064 (0.667) | 0.112 (0.670) | 0.128 (0.673) | 0.137 (0.523) |
| *Refinement* | | | | |
| No. of atoms: protein, ligand, water | 3714, 22, 458 | 3649, 33, 58 | 7481, 12, 278 | 985, 20, 151 |
| R.m.s.d. bond length (Å) | 0.005 | 0.003 | 0.004 | 0.006 |
| R.m.s.d. bond angle (°) | 0.93 | 0.62 | 0.60 | 0.74 |
| Average $B$-factors (Å²): protein, ligand, water | 47.66, 85.08, 49.19 | 69.56, 87.80, 61.55 | 37.39, 38.15, 35.24 | 20.06, 28.71, 35.32 |
| Ramachandran plot (%): favored, allowed, outliers | 99.6, 0.4, 0.0 | 99.1, 0.9, 0.0 | 99.4, 0.6, 0.0 | 99.1, 0.9, 0.0 |
| $R_{work}$, $R_{free}$ | 0.219, 0.245 | 0.220, 0.239 | 0.230, 0.260 | 0.169, 0.207 |

Values in parentheses are for the highest resolution shell

MIU2 to K63-Ub₂ or that to M1-Ub₂ (Table 1, Supplementary Fig. 1c). In contrast, RNF168 UDM2 bound K63-Ub₂ with substantially higher affinity ($K_d = 34\,\mu M$) than M1- or K48-Ub₂ ($K_d = 238\,\mu M$ or $298\,\mu M$, respectively) (Table 1, Supplementary Fig. 1c). These findings suggest that the N- and/or C-terminal regions of UDM2 (i.e., residues 419–435 and/or residues 463–487) (Fig. 1), which are flanked by MIU2, contain essential elements for specific recognition of K63 chains by UDM2. We then examined binding of UDM2 lacking its C-terminal region (UDM2ΔC) to K63-, K48- or M1-Ub₂ by SPR analysis (Table 1, Supplementary Fig. 1c). Although the C-terminal region of UDM2 contains the conserved LRM2 motif, a previous study showed that mutations in this motif have no effect on the interaction between RNF168 and Ub in vivo[20]. Consistently, deletion of the C-terminal region had little effect on specific binding of UDM2 to K63-Ub₂ (Table 1, Supplementary Fig. 1c; we also confirmed that the GST tag did not affect the linkage specificity of UDM2ΔC by comparing the GST-tagged and untagged UDM2ΔC proteins). Our pull-down assay using K6-, K11-, K27-, K29-, K33-, K48-, K63-, and M1-Ub₂ species also suggested that specific binding of UDM2 to K63-Ub₂ does not require its C-terminal region (Supplementary Fig. 1a, b). Therefore, the N-terminal region of UDM2 likely contains an essential element for specific recognition of K63 chains. This region is hereafter referred to as 'Ub-associated domain' (UAD), based on the following structural study (Fig. 1).

**Structure of RNF168 UDM2ΔC in complex with K63-Ub₂.** To confirm the specific interaction between RNF168 UAD–MIU2 and K63 chains and elucidate how UAD contributes to it, we determined the crystal structure of RNF168 UDM2ΔC in complex with K63-Ub₂ at 1.80 Å resolution (Table 2). This structure was determined by molecular replacement using the complex between Rabex-5 MIU and Ub as the search model[31]. The asymmetric unit contains one UDM2ΔC molecule and one of the two Ub moieties in K63-Ub₂ (Supplementary Fig. 4a). UDM2ΔC folds into a single-continuous α-helix (Fig. 3a). UAD and MIU2 in one molecule interact with two distinct Ub moieties that are related by crystallographic symmetry (Fig. 3a, Supplementary Fig. 4a). Although electron density corresponding to the C-terminal three residues (⁷⁴Arg-Gly-Gly⁷⁶) of each Ub moiety is invisible, Leu73 of the UAD-interacting Ub moiety is located in close proximity (10 Å) to Lys63 of the MIU2-interacting Ub moiety (Supplementary Fig. 4a). Therefore, we could unambiguously assign the UAD- and MIU2-interacting Ub moieties as Ub^dist and Ub^prox, respectively, in the context of the complex between RNF168 UDM2ΔC and K63-Ub₂ (Fig. 3a). No observed electron density around the linkage between Gly76 of Ub^dist and Lys63 of Ub^prox may be due to possible high flexibility of the linkage and to the fact that this density corresponds to the average density of the C-terminally linked and non-linked tails of Ub moieties. In this complex, K63-Ub₂ adopts an extended configuration, where Ub^dist and Ub^prox bound the opposite sides of the UDM2ΔC helix.

**Recognition of K63-Ub₂ by UAD–MIU2 of RNF168 UDM2.** In UAD, Leu429, Ile430, Leu432, and Leu436 of RNF168 form a hydrophobic surface to interact with the Ile36-centered hydrophobic patch of Ub^dist, which comprises Ile36, Pro37, Leu71, Leu73 and the aliphatic portion of Thr9 (Fig. 3b, left). This interaction is further stabilized by a hydrogen bond between Glu433 of RNF168 and Thr9 of Ub^dist (Fig. 3b, right). The L436T or E433R mutation of RNF168 decreased the affinity for K63-Ub₂ to 18 or 13% of wild-type affinity, respectively, with little effects on the affinity for M1- or K48-Ub₂ (Table 1, Supplementary Fig. 1c). Both the hydrophobic interaction and hydrogen bond are critical for specific recognition of K63-Ub₂. Glu433 is completely conserved, whereas the residues for the hydrophobic interaction are conserved or replaced by functionally equivalent residues, except for chicken RNF168 (Fig. 3c).

The interaction between RNF168 MIU2 and Ub^prox is similar to that between Rabex-5 MIU and Ub[30, 31]. The hydrophobic surface formed by Leu449, Ala450, and Leu453 of RNF168 interacts with the Ile44-centered hydrophobic patch of Ub^prox

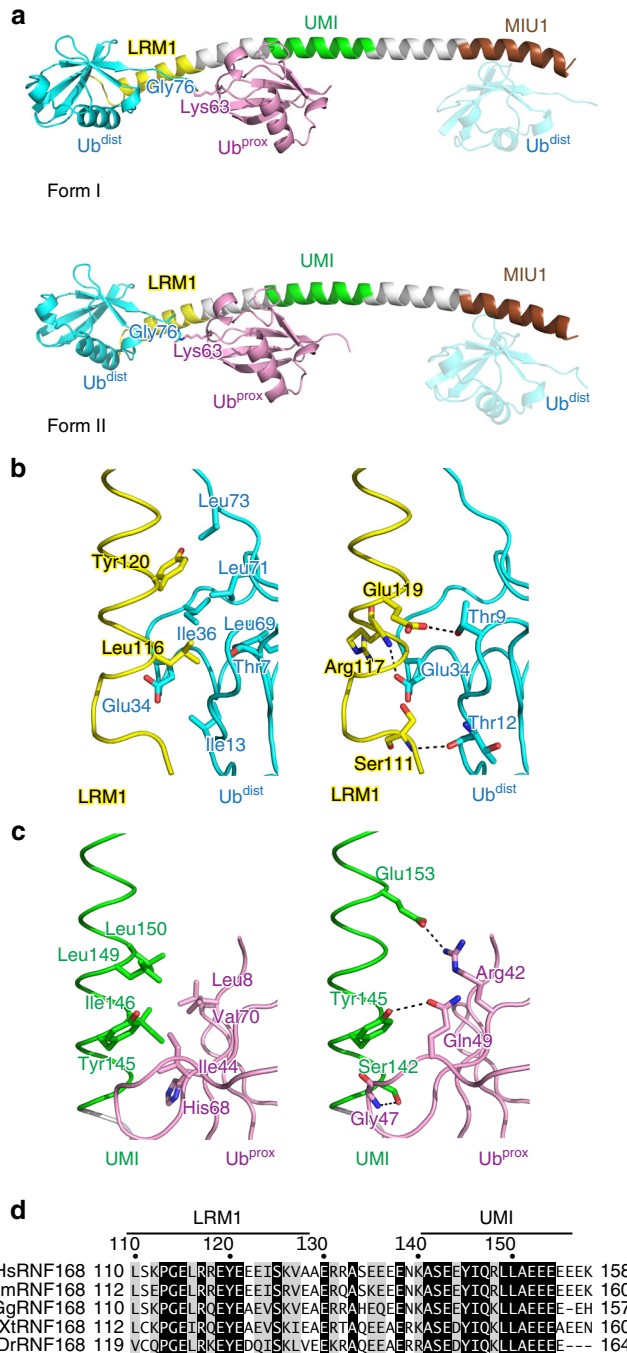

**Fig. 2** Structure of the complex between RNF168 UDM1 and K63-Ub₂. **a** Overall structure of the complex. The coloring scheme of UDM1 is the same as in Fig. 1. Ub$^{dist}$ and Ub$^{prox}$ of K63-Ub₂ are colored cyan and pink, respectively. The linkage between Gly76 of Ub$^{dist}$ and Lys63 of Ub$^{prox}$ is shown. The MIU1-bound Ub$^{dist}$ from the adjacent complex in the crystal is also shown as a translucent model. **b** Hydrophobic (left) and hydrogen-bonding (right) interactions between RNF168 LRM1 and Ub$^{dist}$. The coloring scheme is the same as in **a**, **c** Hydrophobic (left) and hydrogen-bonding (right) interactions between RNF168 UMI and Ub$^{prox}$. The coloring scheme is the same as in **a**, **d** Sequence alignment of LRM1–UMI in human (Hs), mouse (Mm), chicken (Gg), xenopus (Xt), and zebrafish (Dr) RNF168 proteins[56]]. Fully conserved residues are colored white with black background, whereas residues with similar properties (scoring >0.5 in the Gonnet matrix[57]) are marked with gray backgrounds. Asterisks represent the residues whose side chains interact with K63 chains

(Fig. 3d, left). This hydrophobic interaction is further stabilized by hydrogen bonds between the side chains of RNF168 Gln442, Asp446, and Gln454 and the main chains of Ub$^{prox}$ (Fig. 3d, right). The D446A mutation of RNF168 eliminated the binding to M1-, K48-, or K63-Ub₂ (Table 1, Supplementary Fig. 1c). Besides the MIU2-mediated interaction with Ub$^{prox}$, Arg439, which is located between UAD and MIU2, interacts with the main-chain atom of Lys63 of Ub$^{prox}$ (Fig. 3d, right). The R439A mutation of RNF168 decreased the affinity for K63- or M1-Ub₂ to 13 or 44% of wild-type affinity, respectively, and eliminated the binding to K48-Ub₂ (Table 1, Supplementary Fig. 1c). The interactions with Ub$^{prox}$ primarily contribute to the binding affinity to Ub₂ rather than the linkage specificity. Similarly to RNF168 LRM1–UMI, the Lys63-linkage specificity of RNF168 UDM2ΔC depends on the interaction with Ub$^{dist}$ that is linked to Lys63 of Ub$^{prox}$.

**Mechanisms for the specificity of RNF168 UDM1 and UDM2.** Previous structural studies on linkage-specific UBDs have shown that the relative spacing and orientations of their Ub$^{dist}$- and Ub$^{prox}$-interacting surfaces determine the specificities to certain linkage types of Ub chains: the bound Ub$^{dist}$ and Ub$^{prox}$ are fixed on the linkage-specific UBDs and thereby only specific lysine residue(s) and/or Met1 of Ub$^{prox}$ can be physically connected to Gly76 of Ub$^{dist}$[3]. Since the C-terminal tail of Ub$^{dist}$ (residues 71–76) is flexible, the structural mechanism of the linkage specificity of UBDs has been discussed on the basis of the length between Leu71 of Ub$^{dist}$ (the first residue of the C-terminal tail) and lysine residues or Met1 of Ub$^{prox}$ in the UBD-bound diUb structure.

In the UDM1–K63-Ub₂ structure, Ub$^{dist}$ is positioned so that the last β-strand (residues 66–70) of Ub$^{dist}$ is orientated toward the side chain of Lys63 in Ub$^{prox}$ (Fig. 4a). The following C-terminal tail of Ub$^{dist}$ (11-Å length in the apo crystal structure K63-Ub₂[33–35]) is stretched along the UDM1 helix (Fig. 4a, Supplementary Fig. 5a). The Cα–Nε distance between Leu71 of Ub$^{dist}$ and Lys63 of Ub$^{prox}$ varies from 12 to 16 Å in the different complexes in forms I and II (Supplementary Fig. 5a). Although the difference in the Cα–Nε distance (4 Å) suggests that the interdomain region between LRM1 and UMI tolerates some flexibility, the positions of Ub$^{dist}$ and Ub$^{prox}$ in the UDM1-bound K63-Ub₂ seem incompatible with the binding to other Ub-chain types. Apart from Lys63, the lysine residues of Ub$^{prox}$ are located 29- to 39-Å apart from Leu71 of Ub$^{dist}$ (Supplementary Fig. 5b). These distances are longer than the length of the C-terminal tail of Ub$^{dist}$ at full extension (~20 Å). Therefore, the co-operative binding between LRM1–UMI and Ub₂ seems possible only for K63-Ub₂, unless the α-helix conformation of LRM1–UMI is distorted. Although both Lys63 and Met1 are close to the Ub$^{dist}$ C-terminal tail in the apo form, the orientation of Ub$^{dist}$ in the UDM1–K63-Ub₂ complex shifts Met1 22 Å away from Leu71 of Ub$^{dist}$ (Fig. 4a). Therefore, the distance between LRM1 and UMI, as well as the respective orientations of the LRM1–UMI-bound Ub moieties are responsible for the specific interaction between LRM1–UMI and K63-Ub₂. Note that the Cα–Nε distance between Leu71 of the UMI-bound Ub moiety and Lys63 of the MIU1-bound Ub moiety in the crystals of the UDM1–K63-Ub₂ complex is estimated to be 24 Å in form I (Fig. 4a; Lys63 is mutated to Arg63 in this structure), which is longer than the Cα–Nε distance of K63-Ub₂ at full extension (~20 Å). K63-Ub₂ cannot simultaneously interact with both UMI and MIU1, in contrast to the binding with LRM1–UMI.

In the complex between RNF168 UDM2ΔC and K63-Ub₂, the last β-strand of Ub$^{dist}$ is also directed toward Lys63 of Ub$^{prox}$ (Fig. 4b). In this structure, the interactions of UAD and MIU2 with the Ub moieties occur at the opposite sides of the UDM2ΔC

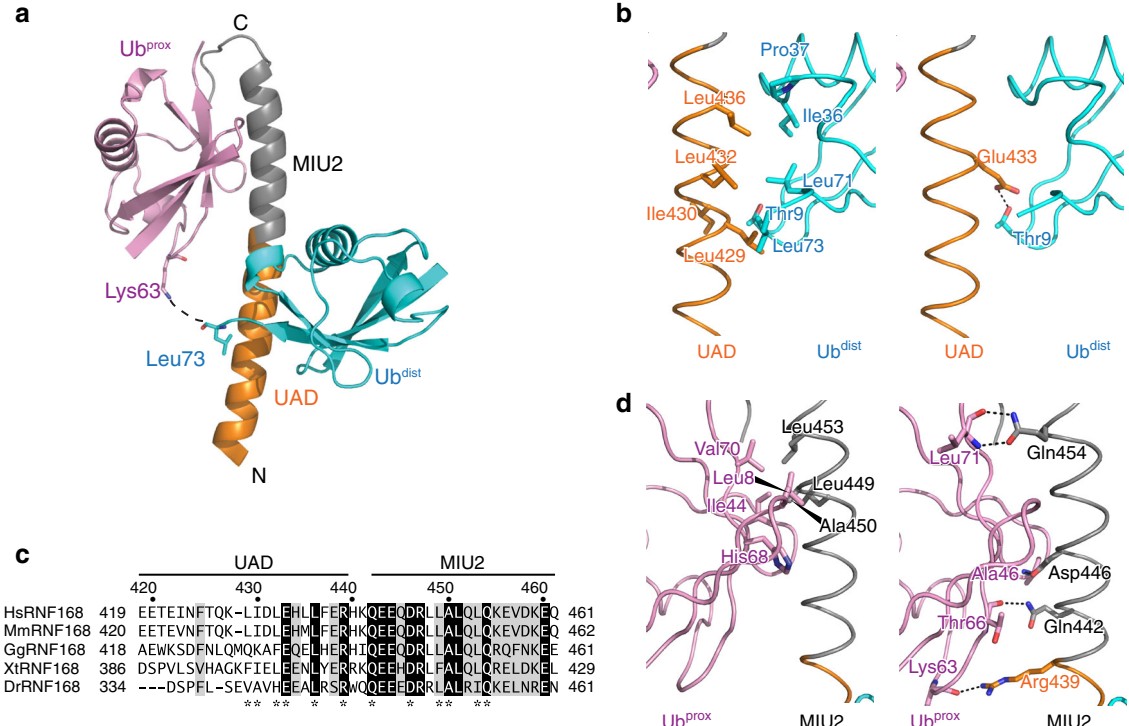

**Fig. 3** Structure of the complex between RNF168 UDM2ΔC and K63-Ub$_2$. **a** Overall structure of the complex. The coloring scheme of UDM2 is the same as in Fig. 1. Ub$^{dist}$ and Ub$^{prox}$ of K63-Ub$_2$ are colored cyan and pink, respectively. The expected linkage between Ub$^{dist}$ and Ub$^{prox}$ is drawn as a dotted line. **b** Hydrophobic (left) and hydrogen-bonding (right) interactions between RNF168 UAD and Ub$^{dist}$. The coloring scheme is the same as in (**a**), **c** Sequence alignment of UDM2ΔC in human (Hs), mouse (Mm), chicken (Gg), xenopus (Xt), and zebrafish (Dr) RNF168 proteins[56]. The drawing scheme is the same as in Fig. 2d. **d** Hydrophobic (left) and hydrogen-bonding (right) interactions between RNF168 MIU2 and Ub$^{prox}$. The coloring scheme is the same as in (**a**)

helix (Fig. 4b). As a result, the C-terminal tail of Ub$^{dist}$ that is linked to Lys63 of Ub$^{prox}$ may wrap around the α helix. The Cα–Nε distance between Leu71 of Ub$^{dist}$ and Lys63 of Ub$^{prox}$ is 15 Å, which is substantially shorter than those between Leu71 of Ub$^{dist}$ and the other lysine residues of Ub$^{prox}$ or the Cα–N distance between Leu71 of Ub$^{dist}$ and Met1 of Ub$^{prox}$ (Fig. 4b, Supplementary Fig. 5c). Although the Cα–Nε distance between Leu71 of Ub$^{dist}$ and Lys6 of Ub$^{prox}$ is 17 Å, the Ile36–Glu41 region of Ub$^{dist}$ is located between the two residues. Such a Ub$^{dist}$–Ub$^{prox}$ connection needs to get around this region (Supplementary Fig. 5c). Collectively, the spacing between the Ub$^{dist}$-interacting region of UAD and the Ub$^{prox}$-interacting region of MIU2 determines the specificity of UDM2 for K63 chains.

**UDM2-dependent recruitment of RNF168 in U2OS cells**. Our present studies on the structures and interactions of UDM1 and UDM2 with K63-Ub$_2$ confirmed their binding specificities for K63 chains in vitro. To dissect their functional roles in the context of the DSB response, we analyzed the accumulation of RNF168 at DSBs induced by gamma irradiation (Fig. 5). First, we established U2OS cells that harbor tetracycline-inducible TagBFP-P2A-siRNA-resistant RNF168 transgene. We sorted TagBFP-positive cells after doxycycline treatment and used the heterogeneous population of the cells. One of two different siRNF168-specific siRNAs (siRNF168#C and siRNF168#5) was used for silencing endogenous RNF168 expression (Fig. 5, Supplementary Fig. 6). When control siRNA-transfected U2OS cells were treated with gamma irradiation, the endogenous RNF168 formed ionizing radiation induced foci (IRIF). Similarly, wild-type RNF168, whose expression was induced by doxycycline,

formed IRIF in siRNF168#C-transfected U2OS cells (Fig. 5a). These RNF168 foci co-localized with γH2AX foci (Fig. 5a).

We then analyzed the foci formation of different RNF168 mutants (Fig. 5). We confirmed that the expression of these mutants could be induced by doxycycline treatment in U2OS cells (Supplementary Fig. 6a, b). Mutations that impair binding of RNF168 UDM1 to K63-Ub$_2$ (L116A in LRM1 and D175A in MIU1) had little effect on the foci formation (95 and 80% of wild type, respectively), whereas the S142A mutation in UMI mildly affected the foci formation (64% of wild type, $p$-value < 0.0001). Therefore, binding of UDM1 to K63 chains is dispensable for the RNF168 accumulation at DSB sites. In previous experiments using 10 Gy of gamma irradiation instead of 3 Gy, a truncation of the whole MIU1 (Δ168–191) did not affect the RNF168 accumulation, whereas that of UMI (Δ134–165) only mildly affected it[20]. Overall, the interaction of UDM1 with Ub seems negligible to recruit RNF168 to DSB sites.

The D446A mutation in MIU2 of UDM2, which eliminates binding of RNF168 UDM2ΔC to Ub$_2$, strongly affected the number of U2OS cells with RNF168 foci (1% of wild type, $p$-value < 0.0001). The MIU2-mediated interaction with K63 chains is critical for the RNF168 accumulation. This finding is consistent with a previous observation that the A450G mutation in MIU2 of UDM2 also impaired the RNF168 foci formation[20]. The E433R or R439A mutation in UAD of UDM2, which decreased the affinity for K63-Ub$_2$ to 13 or 7% of wild-type affinity, respectively, had intermediate effects (62 or 45% of wild type, $p$-value < 0.0001). The UAD-mediated interaction is important but less than the MIU2-mediated interaction. The combination of the UMI mutation (S142A) and UDM2 mutation (E433R or R439A) had a synergistic effect on the RNF168 accumulation. UMI contributes to the RNF168 accumulation, especially in the absence of

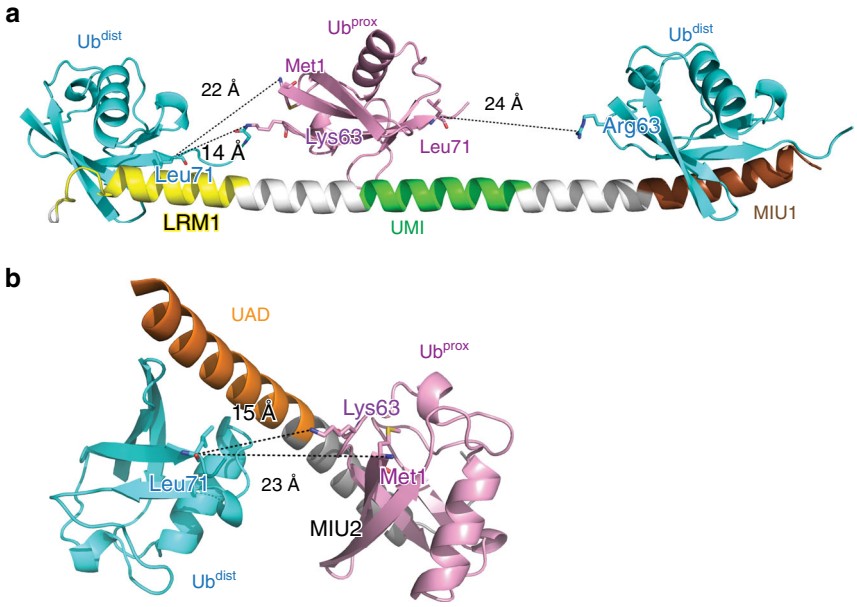

**Fig. 4** Spacing between $Ub^{dist}$ and $Ub^{prox}$ of $K63\text{-}Ub_2$ bound to UDM1 or UDM2ΔC of RNF168. **a** Spacing between $Ub^{dist}$ and $Ub^{prox}$ of $K63\text{-}Ub_2$ bound to UDM1 of RNF168 in the form I crystal. The coloring scheme is the same as in Fig. 2. **b** Spacing between $Ub^{dist}$ and $Ub^{prox}$ of $K63\text{-}Ub_2$ bound to UDM2ΔC of RNF168. Same coloring scheme as in Fig. 3

the functional UDM2. In contrast, either LRM1 or MIU1 mutation had little or no effect on the RNF168 accumulation, even in the combination with the R439A mutation in MIU2 (Fig. 5b). The experiments using another siRNA (siRNF168#5) provided similar results (Supplementary Fig. 6).

## Discussion

The UMI structure presented in this study is similar to the structures of other α-helical UBDs including UIMs, MIUs and the α-helix in FAAP20 UBZ[30, 31, 36–38] (Supplementary Fig. 7a), although the direction of the Ub-interacting α-helix in UIMs is reversed to those in the others. However, the UMI–Ub interaction is slightly different from the interactions between these α-helical UBDs and Ub. The UMI of RNF168 lacks a critical Ala that is conserved among other α-helical UBDs for the interaction with the hydrophobic pocket formed by Leu8, Val70 and Ile44 of Ub. In RNF168 UMI, this Ala is replaced by Ile, which tilts the α-helix in regard to Ub (Supplementary Fig. 7a). Despite this difference, UMI, MIU1, and MIU2 of RNF168 have similar affinities for monoUb. Both MIU1 and MIU2 of RNF168 are classified as a canonical, non-linkage-specific MIU, and are different from the recently found K48-linkage-specific MIU of the deubiquitylase MINDY-1, which can form multiple Ub-interacting sites for the recognition of Lys48-linked triUb[39]. In UDM1 and UDM2ΔC, the simultaneous interactions with $Ub^{dist}$ and $Ub^{prox}$ provide the specificity for K63 chains. Similar avidity mechanisms have also been found in other linkage-specific Ub receptors, such as RAP80 and Epsin1[5, 25, 27], which contain two equivalent UBDs in terms of the affinity to monoUb. In contrast, one of the two UBDs in UDM1 and UDM2ΔC (i.e., LRM1 and UAD, respectively) has no obvious Ub-binding ability in the absence of their associated UBDs (i.e., UMI and MIU2 for UDM1 and UDM2ΔC, respectively). Such 'weak' Ub-interacting elements for linkage-specific binding to Ub chains might be hidden in the adjacent regions of known single UBDs in Ub receptors.

RNF169 is a paralog of RNF168 and also contains both UDM1 and UDM2 (Supplementary Fig. 7b). However, it has been reported that RNF169 equally binds K63 and K48 chains[20–22]. In UDM1, most of the Ub-interacting residues are conserved in LRM1 and UMI of RNF169, whereas the length of the inter-domain region between LRM1 and UMI, which is critical for specific recognition of K63 chains, is different between RNF168 and RNF169 (Supplementary Fig. 7b). Consistently, the replacement of the interdomain region of RNF168 UDM1 by that of RNF169 UDM1 abrogated the specific interaction with $K63\text{-}Ub_2$ in vitro (Supplementary Fig. 7c). In UDM2, the Ub-interacting residues of UAD are completely different between RNF168 and RNF169 (Supplementary Fig. 7b). These differences are consistent with the previous finding that binding of RNF169 to Ub chains is not specific to K63 chains in vitro and may explain why RNF169 cannot be recruited directly by RNF8-mediated polyubiquityla-tion in U2OS cells, in contrast to RNF168[20–22].

A previous study showed that the truncation of LRM1 has a strong impact on the RNF168 recruitment[20]. Specifically, the removal of residues 110–113 affected the RNF168 foci formation. However, this region is not involved in binding to Ub in our structure. On the contrary, a mutation of LRM1 (L116A), which affects the interaction between LRM1–UMI and $K63\text{-}Ub_2$, had no measurable effect on the RNF168 recruitment. Therefore, LRM1 may have a Ub-independant function on the RNF168 recruitment in vivo. Originally, LRM1 has been suggested to interact directly with polyubiquitylated targets[13]. Consistently, a peptide encompassing LRM1 can bind to linker histone H1, which becomes polyubiquitylated upon inducing DSBs[13]. However, the direction of Ub relative to UMI indicates that LRM1 is probably far from the polyubiquitylated targets, which should be facing the C-terminal part of UDM1. Further studies on the interaction between histone H1 and RNF168 are necessary to understand the Ub-independent function of LRM1. Since mutations in UAD are epistatic with the inactivation of UMI, UDM2ΔC may also be involved in the recognition of K63 chains during DNA damage response. Although UDM1 and UDM2 are distant in the primary sequence of RNF168 (Fig. 1), these domains may be three-dimensionally close to each other so that they can bind to the same targets.

The catalytic RING domain of RNF8 enhances the assembly of K63 chains by Ubc13–Mms2 in vitro, whereas that of RNF168 has little effect on it[12]. On the other hand, several studies using the full-length RNF168 showed that it facilitates the assembly of

K63 chains on H2A and H2AX in cooperation with RNF8 in vitro[17, 40] and in cells[14, 15] in RING- and UDM2-dependent manners. Therefore, in RNF168, UDM2 may play important roles not only in the recruitment to DSB sites but also in the assembly of K63 chains on H2A and H2AX. MIU2 but not LRM2 is required for binding to K63 chains, although LRM2 is critical for the accumulation of RNF168. Our present study provided a structural basis on the functional role of UDM2 in the recruitment of RNF168: UAD–MIU2 can simultaneously interact with Ub$^{prox}$ and Ub$^{dist}$ of K63-Ub$_2$ and specifically recognizes K63 chains without LRM2. The geometry between UAD–MIU2 and LRM2 raise the possibility that LRM2 could interact with H2A or H2AX. Further structural and functional studies on the relationship with (poly)ubiquitylated targets are awaited for complete understanding of the hierarchical signaling mechanism mediated by DSB-responsive factors including RNF8, RNF168, and RNF169.

## Methods

**Preparation of RNF168 domains**. UDM1 (residues 110–188 or 113–188), LRM1 (residues 110–137), UMI (residues 134–166), LRM1–UMI (residues 110–166), UAD (residues 419–437), MIU2 (residues 436–462), UDM2ΔC (residues 419–462), and UDM2 (residues 419–487) of human RNF168 were cloned into pCold-GST vectors using NdeI and XhoI. Primer sequences used in this study are shown in Supplementary Data 1. Escherichia coli Rosetta (DE3) cells (Invitrogen) were transformed with the individual expression vector. The transformed cells were cultured in LB medium containing 100 mg L$^{-1}$ ampicillin at 37 °C. At an optical density (600 nm) of ~0.5, isopropyl-β-D-thiogalactopyranoside (IPTG) was supplemented to a final concentration of 0.1 mM. The culture for protein expression was continued for 24 h at 15 °C. The cells were collected by centrifugation at 7000×g for 15 min and disrupted by sonication in phosphate buffered saline (PBS) containing 1 mM dithiothreitol (DTT) and 0.5% Triton X-100. The lysates were cleared by centrifugation at 28,000×g for 60 min. The supernatants were then loaded onto a Glutathione Sepharose FF column(GE Healthcare) pre-equilibrated with PBS containing 1 mM DTT and 0.5% Triton X-100. The column was washed with PBS containing 1 mM DTT and 0.5% Triton X-100 and then with PBS containing 1 mM DTT. The GST-fused proteins were eluted with 50 mM Tris-HCl buffer (pH 8.0) containing 200 mM NaCl, 1 mM DTT and 15 mM reduced glutathione.

For UDM1, the GST tag was cleaved by HRV3C protease at 4 °C overnight. The protease-treated UDM1 sample was then loaded onto a HiLoad 16/60 Superdex75 size-exclusion column (GE Healthcare) with 20 mM Tris-HCl buffer (pH 8.0) containing 50 mM NaCl and 5 mM β-mercaptoethanol. For UDM2ΔC, the sample was dialyzed against 50 mM Tris-HCl buffer (pH 8.0) containing 1 mM DTT in the presence of HRV3C protease to cleave the GST tag. The UDM2ΔC sample was loaded onto a ResourceQ anion exchange column (GE Healthcare) pre-equilibrated with 50 mM Tris-HCl buffer (pH 8.0) containing 1 mM DTT and then eluted with a linear gradient of 0–1 M NaCl. The peak fractions containing RNF168 UDM2ΔC were collected, concentrated and loaded onto a HiLoad 16/60 Superdex75 size-exclusion column (GE Healthcare) with 20 mM Tris-HCl buffer (pH 8.0) containing 50 mM NaCl and 5 mM β-mercaptoethanol. The fractions abundant in the purified UDM1 or UDM2ΔC protein were collected and concentrated to ~10 g L$^{-1}$ with an Amicon Ultra-4 10,000 MWCO filter (Millipore). Circular dichroism (CD) spectra of wild type and mutant proteins were measured to confirm that the mutations did not affect the overall structure of the RNF168 domains (Supplementary Fig. 8).

For His$_6$-tagged proteins, the cells were disrupted by sonication in PBS containing 20 mM imidazole and 0.5% Triton X-100. The lysates were centrifuged at 28,000×g for 1 h. The supernatants were then loaded onto a Ni-NTA (Qiagen) column pre-equilibrated with PBS containing 20 mM imidazole and 0.5% Triton X-100. The column was washed with PBS containing 20 mM imidazole and 0.5% Triton X-100 and then with PBS containing 20 mM imidazole. The His$_6$-tagged proteins were eluted with PBS containing 200 mM imidazole and then dialyzed against 50 mM Tris-HCl buffer (pH 8.0) containing 1 mM DTT. The samples were then purified in a manner similar to UDM2ΔC.

**Fig. 5** Foci formation of wild-type or mutant RNF168 after gamma irradiation. U2OS cells that harbor tetracycline-inducible siRNA-resistant RNF168 were first transfected with RNF168 siRNA (siRNF168#C) or non-targeting control siRNA (siCTRL). Expression of siRNA-resistant RNF168 (RNF168*) was then induced by doxycycline. The cells were irradiated (3 Gy) and processed for immunostaining using RNF168 and γH2AX antibodies. **a** Representative images of immunofluorescence are shown. The nuclei visualized by DAPI staining were outlined with white borders. The scale bars indicate 5 μm. Green: RNF168, Red: γH2AX. **b** Quantitation of **a**. 90 cells were analyzed for each sample. The line represents the mean of the number of foci per cell. Significance is reported as the Kruskal–Wallis test (*p < 0.05; **p < 0.01; ***p < 0.001)

**Preparation of Ub$_2$ species**. Ub or M1-Ub$_2$ was overproduced at 20 °C in *E. coli* strain Rosetta (DE3) cells (Invitrogen) transformed with the pET26b (Novagen) expression vector harboring mouse Ub or human M1-Ub$_2$ gene, respectively[41]. The cells were disrupted by sonication in 50 mM ammonium acetate buffer (pH 4.5) containing 200 mM NaCl. The cleared lysates were incubated at 80 °C for 5 min. The denatured and insolubilized *E. coli* proteins were pelleted by centrifugation at 30,000×*g* for 30 min. The supernatants were dialyzed against 50 mM ammonium acetate buffer (pH 4.5). The samples were purified by a ResourceS cation exchange column (GE Healthcare) and a HiLoad 16/60 or 26/60 Superdex75 size-exclusion column (GE Healthcare). The purified Ub and M1-Ub$_2$ were concentrated with an Amicon Ultra-15 10,000 MWCO filter (Millipore).

K6-, K11-, K29-, K33-, K48-, and K63-Ub$_2$ samples were synthesized enzymatically. For K6-Ub$_2$ synthesis, E1 (0.3 μM), UbcH7 (8 μM), NleL (2.5 μM), Ub (1.2 mM), and OTUB1 (10 μM) were mixed in PBDM buffer (50 mM Tris-HCl buffer (pH 7.6) containing 2 mM ATP, 2 mM DTT, 5 mM MgCl$_2$, 10 mM creatine phosphate (Sigma-Aldrich), 0.6 U mL$^{-1}$ of creatine phosphokinase (Sigma-Aldrich) and 0.6 U mL$^{-1}$ of inorganic pyrophosphatase (Sigma-Aldrich)) and incubated at 37 °C overnight[42, 43]. For K11-Ub$_2$ synthesis, E1 (0.3 μM), E2S-UBP (40 μM), Ub (1.2 mM), and AMSH-LP (0.8 μM) were mixed in PBDM buffer and incubated at 30 °C for 30 h[44]. After 10 h of incubation, AMSH-LP (2 μM) was further supplemented. For K29-Ub$_2$ synthesis, E1 (0.6 μM), UbcH5c (8 μM), E3C (6 μM), and Ub (1.2 mM) were mixed in PBDM buffer containing 10% glycerol and incubated at 30 °C for 30 h[45]. After 10 h of incubation, OTUB1 (5 μM), AMSH-LP (1 μM), and Cezanne (E287K, E288K; 20 μM) were further supplemented. For K33-Ub$_2$ synthesis, E1 (0.6 μM), UbcH7 (8 μM), AREL (6 μM) and Ub (1.2 mM) were mixed in PBDM buffer containing 10% glycerol and incubated at 30 °C for 30 h[45, 46]. After 10 h of incubation, OTUB1 (5 μM), AMSH-LP (1 μM) and Cezanne (E287K, E288K; 20 μM) were further supplemented. For K48-Ub$_2$ synthesis, E1 (0.3 μM), E2-25K (8 μM) and Ub (1.2 mM) were mixed in PBDM buffer and incubated at 37 °C overnight[47]. For K63-Ub$_2$ synthesis, E1 (0.3 μM), Ubc13 (8 μM), MMS2 (8 μM) and Ub (1.2 mM) were mixed in PBDM buffer and incubated at 37 °C overnight[48]. Each reaction solution was mixed with five volumes of 50 mM ammonium acetate buffer (pH 4.5) and loaded onto a ResourceS cation exchange column (GE Healthcare) pre-equilibrated with 50 mM ammonium acetate (pH 4.5) buffer containing 100 mM NaCl. The synthesized Ub$_2$ species were eluted with a linear gradient of 100–500 mM NaCl in 50 mM ammonium acetate buffer (pH 4.5). Peak fractions containing Ub$_2$ were loaded onto a HiLoad 16/60 Superdex75 size-exclusion column (GE Healthcare) with 10 mM Tris-HCl buffer (pH 7.2) containing 50 mM NaCl and 5 mM β-mercaptoethanol.

The purified Ub$_2$ species were concentrated to ~30 g L$^{-1}$ and stored at −80 °C until use. K27-Ub$_2$ was purchased from LifeSensors (Cat. #SI2702).

**Crystallization and data collection**. To prepare the complex between RNF168 UDM1 and K63-Ub$_2$, a 1.5-fold molar excess of RNF168 UDM1 was incubated for 1 h at 4 °C with K63-Ub$_2$. For crystallization, K63-Ub$_2$ was synthesized from an equimolar mix of K63R Ub and D77 Ub instead of wild-type Ub. Both K63R Ub and D77 Ub were prepared in a manner similar to wild-type Ub. To prepare the complex between RNF168 UDM2ΔC and K63-Ub$_2$, a twofold molar excess of RNF168 UDM2ΔC was incubated for 1 h at 4 °C with K63-Ub$_2$. The complexes were loaded onto a HiLoad 16/60 Superdex75 size-exclusion column (GE Healthcare) with 20 mM Tris-HCl buffer (pH 8.0) containing 50 mM NaCl and 5 mM β-mercaptoethanol to remove the unbound UDM1 or UDM2ΔC. The purified complexes were concentrated to ~10 g L$^{-1}$ by using an Amicon Ultra-4 10,000 MWCO filter (Millipore). Initial crystallization screening by a Mosquito liquid-handling robot (TTP Lab Tech) was carried out at 20 °C in 96-well sitting drop vapor diffusion plates. About 600 conditions were tested, using crystallization reagent kits supplied by Hampton Research. Crystallization conditions found by the initial screening were further optimized.

Form I crystals of the complex between RNF168 UDM1 and K63-Ub$_2$ were grown at 20 °C with the sitting drop vapor diffusion method by mixing 1 μL of the protein solution with an equal amount of precipitant solution containing 15% PEG3350, 0.1 M Tris-HCl (pH 8.5) and 100 mM MgCl$_2$. For data collection, the form I crystals were transferred to 15% PEG3350, 0.1 M Tris-HCl (pH 8.5), 100 mM MgCl$_2$ and 21% xylitol for cryoprotection. Form II crystals of the complex between RNF168 UDM1 and K63-Ub$_2$ were grown at 20 °C with the hanging drop vapor diffusion method by mixing 1 μL of the protein solution with an equal amount of precipitant solution containing 23% PEG MME 2000, 0.1 M Bis-Tris-HCl (pH 6.5) and 10 mM Pr acetate. For data collection, the form II crystals were transferred to 23% PEG MME 2000, 0.1 M Bis-Tris-HCl (pH 6.5) and 30% glycerol for cryoprotection. Tetrameric form crystals of the complex between RNF168 UDM1 and K63-Ub$_2$ were grown at 20 °C with the hanging drop vapor diffusion method by mixing 1 μL of the protein solution with an equal amount of precipitant solution containing 21% PEG3350, 0.1 M Tris-HCl (pH 7.6). For data collection, the tetrameric form crystals were transferred to 21% PEG3350, 0.1 M Tris-HCl (pH 7.6) and 30% glycerol for cryoprotection. Crystals of the complex between RNF168 UDM2ΔC and K63-Ub$_2$ were grown at 20 °C with the sitting drop vapor diffusion method by mixing 0.2 μL of the protein solution with an equal amount of precipitant solution containing 25% PEG33500, 1 M Bis-Tris-HCl (pH 6.5) and 0.2 M ammonium acetate. For data collection, the crystals of the complex between RNF168 UDM2ΔC and K63-Ub$_2$ were transferred to 25% PEG3350, 0.1 M Bis-

Tris-HCl, (pH 6.5), 0.2 M ammonium acetate and 30% ethylene glycol for cryoprotection. The cryoprotected crystals were flash frozen in liquid nitrogen.

**Structural determination and refinement**. The diffraction data were collected at 100 K at the beamline BL41XU in SPring-8 (Hyogo, Japan) and then processed using the program HKL2000[49] and the CCP4 program suite[50]. All complex structures presented in this study were determined by molecular replacement using the program MolRep[51]. The crystal structure of the Rabex-5 MIU–Ub complex (PDB 2FID) was used as the search model. The atomic models were built to fit $2F_o−F_c$ electron density map by using the program COOT[52]. Structure refinement was carried out by using the program Phenix[53]. The final models obtained after iterative correction and refinement have excellent stereochemistry with $R_{free}$ values of 24.5% at 1.78 Å resolution for the form I complex between RNF168 UDM1 and K63-Ub$_2$, 23.9% at 2.25 Å resolution for the form II complex between RNF168 UDM1 and K63-Ub$_2$, 26.0% at 2.50 Å resolution for the tetrameric complex between RNF168 UDM1 and K63-Ub$_2$, and 20.7% at 1.80 Å resolution for the complex between RNF168 UDM2ΔC and K63-Ub$_2$. The data collection and refinement statistics are shown in Table 2. All molecular graphics were prepared with PyMOL (DeLano Scientific; http://www.pymol.org).

**Pull-down assays**. Quantity of 100 μg of the His$_6$-fused RNF168 samples was immobilized on Ni-NTA beads pre-equilibrated in a pull-down buffer (25 mM Tris-HCl (pH 8.0), 20 mM imidazole, 0.1% Triton X-100, 100 mM NaCl) and then incubated with 15 μg or 1.5 μg Ub$_2$ species in the pull-down buffer for the experiments corresponding to Supplementary Fig. 1a, b, respectively, on ice for 2 h. The beads were extensively washed with the pull-down buffer thrice. The Ub$_2$ molecules bound to the beads were released in SDS loading buffer without boiling and analyzed by SDS-PAGE with Coomassie brilliant blue staining.

**SPR analysis**. SPR measurements were carried out on a Biacore T200 instrument (GE Healthcare) with HBS-P buffer (10 mM HEPES-Na [pH 7.4], 150 mM NaCl and 0.05% surfactant P20) at 25 °C. Anti-GST antibodies (GE Healthcare) were covalently immobilized on the CM5 sensor chip (GE Healthcare) at a density of about 13,000 resonance units (RU). The GST-fused RNF168 domains were then captured on the sensor chip at a density of 1000–1500 RU. The untagged LRM1–UMI and UDM2ΔC were covalently immobilized on the CM5 sensor chip using amine coupling in 10 mM sodium acetate (pH 4.0 and pH 4.5, respectively). Ub$_2$ species were injected for 60 s at a flow rate of 10 μL per min. Equilibrium dissociation constants ($K_d$) were computed by fitting steady-state binding level ($R_{eq}$) to a 1:1 interaction model ($R_{eq}$ = Concentration × $R_{max}/(K_d$ + Concentration) + Offset) using Biacore T200 evaluation software (GE Healthcare). All assays were carried out thrice for each sample. The data are presented as mean ± standard deviation.

**CD measurements**. CD spectra were measured in 10 mM sodium phosphate buffer (pH 7.5) at 25 °C with a J-750 CD spectrophotometer (JASCO) equipped with a quartz cuvette of a 1.0-mm light path. The concentrations of the samples were adjusted to 0.1 g L$^{-1}$. Four scans from 250 to 190 nm were averaged. The data were processed using the Spectra Manager software (JASCO).

**SEC-MALS analysis of UDM1–K63-Ub$_2$**. Volume of 100 μl of UDM1, K63-Ub$_2$ or a 2:1 mixture of UDM1 with K63-Ub$_2$ at a concentration of 1 g L$^{-1}$ was loaded onto an ENrich SEC 650 gel-filtration column (Bio-Rad) in 20 mM Tris-HCl (pH 8.0) buffer containing 50 mM NaCl at 20 °C, online with a DAWN 8+ light scattering detector (Wyatt Technology). The data collection and analysis were carried out using the ASTRA6 software package (Wyatt Technology).

**Human cell lines**. To establish human cells carrying a tetracycline-inducible exogenous *RNF168* gene[54], we used the Retroviral Tet-On 3G Inducible Expression System (Clontech). A nucleotide fragment consisting of a Kozak sequence (5′-CGCCACC-3′) and an ORF encoding a TagBFP-P2A-RNF168 fusion protein, including silent mutations in siRNA-target regions of RNF168, was cloned into *Cla*I and *Not*I restriction sites of pRetroX-TRE3G Vector (Clontech). The silent mutations introduced into *RNF168* gene were as follows: c.495A > G, c.498A > G, c.501G > C, c.828A > C, c.831T > G, and c.834T > C. To generate a series of RNF168 mutants, we further introduced the following individual mutations: L116A (CTG to GCC), S142A (AGT to GCT), D175A (GAT to GCT), E433R (GAG to AGG), R439A (AGA to GCA), and D446A (GAC to GCC). GP2–293 retrovirus packaging cells were transfected with one of these plasmids together with pAmpho envelop vector (Clontech) using Xfect Transfection Reagent (Clontech) according to the manufacturer's instruction. After 48 h of transfection, the supernatant containing recombinant retroviruses was filtered and transduced into human osteosarcoma U2OS Tet-On cells (Clontech) using Polybrene transfection reagent (Millipore). The cells were further cultured overnight, and then selected with puromycin (Nacalai) at a final concentration of 0.5 μg mL$^{-1}$. After puromycine selection, the cells were treated with 0.2 μg mL$^{-1}$ doxycycline and then TagBFP-expressing cells were sorted by FACSAria II (BD).

**Immunofluorescence microscopy analysis.** A total of $1 \times 10^6$ cells were seeded on MAS-coated coverslips (Matsunami Glass). A double-stranded siRNA targeting endogenous RNF168 [either siRNF168#5 (5′-GACACUUUCUCCACAGAUA-UU-3′) or siRNF168#C (5′-GGCGAAGAGCGAUGGAAGA-dTdT-3′)[55]] or a non-targeting control siRNA (Sigma-Aldrich, MISSION siRNA Universal Negative Control #2) was transfected into the cells using Lipofectamine RNAi MAX (Invitrogen). After the cells were cultured overnight, expression of exogenous RNF168 was induced by addition of doxycycline to the medium at a final concentration of $0.2 \ \mu g \ mL^{-1}$ for 2 days. The effects of gene silencing of endogenous RNF168 and expression of exogenous RNF168 were confirmed by western blotting analysis using RNF168 antibody (Millipore). The cells were irradiated with 3 Gy of gamma rays (Gammacell 40 Exactor, $^{137}Cs$, MDS Nordion) to induce double-strand DNA breaks. 30 min after irradiation, the cells were washed twice with PBS and pre-extracted with CSK buffer (20 mM HEPES (pH 7.4), 50 mM NaCl, 3 mM MgCl$_2$, 300 mM sucrose) supplemented with 0.5% Triton X-100 on ice for 5 min. After removal of the CSK buffer, the cells were fixed with 3% paraformaldehyde and 2% sucrose in PBS for 15 min at room temperature, and then washed thrice with PBS followed by blocking with 1% BSA in PBS for 1 h. The cells were stained with rabbit-derived anti-RNF168 (Millipore, #ABE367, dilution 1:100) and mouse-derived anti-γH2AX (Millipore, clone JBW301, dilution 1:2000) antibodies overnight at 4 °C. The cells were washed thrice with 1% BSA in PBS, and then stained with anti-rabbit IgG (H + L) antibody conjugated to Alexa Fluor 488 (Invitrogen, #A11034, dilution 1:1000) and anti-mouse IgG (H + L) antibody conjugated to Alexa Fluor 555 (Invitrogen, #A21424, dilution 1:1000) for 1 h. After the cells were washed twice with PBS, DNA was stained with DAPI (Wako, dilution 1:100,000). The samples were mounted in Prolong Gold antifading reagent (Invitrogen) before microscopy analysis. Fluorescent images were obtained using a fluorescent microscope (Leica, DMI6000 B) and a confocal laser microscope (Leica, TCS SP5). All the images were processed by Adobe Photoshop CS6 software (Adobe Systems). The number of RNF168 foci in each cell was counted using a fluorescent microscope (Leica, DMI6000 B). Cells expressing high level of RNF168 (Supplementary Fig. 6d) were excluded from analysis. We analyzed 30 cells per sample and performed the experiment thrice. For statistical analysis, the Kruskal–Wallis test was performed using the KaleidaGraph software.

**Western blotting.** A double-stranded siRNA targeting endogenous RNF168 or a control siRNA was transfected into cells harboring the tetracycline-inducible siRNA-resistant *RNF168* gene. After the cells were cultured overnight, doxycycline was added to the medium at a concentration of $0.2 \ \mu g \ mL^{-1}$. The cells were further cultured for 2 days. The cultured cells were lysed in SDS sample buffer with boiling for 5 min. The total cell lysate was separated by SDS-PAGE and transferred to nitrocellulose membrane using the iBlot Gel Transfer System (Invitrogen). The membrane was blocked with 5% skim milk (Nacalai) in Tris buffered saline with 0.05% Tween 20, stained with primary antibodies, washed and then stained with horseradish peroxidase (HRP)-conjugated secondary antibodies[55]. Anti-RNF168 antibody (Millipore, #ABE367; dilution 1:500) or anti-α-tubulin antibody (SIGMA, T6074; dilution 1:2000) was used as primary antibody and HRP-conjugated anti-rabbit-IgG antibody (Promega, W401B; dilution 1:2000) or anti-mouse-IgG antibody (Promega, W402B; dilution 1:2000) was used as secondary antibody. Western Lightning Ultra reagent (PerkinElmer Sciences) or Western Lightning ECL Pro reagent (PerkinElmer Sciences) was used to detect HRP activity. Chemiluminescent images were captured by an ImageQuant LAS4000mini lumino-image analyzer (GE Healthcare). Uncropped images of gels and blots are shown in Supplementary Fig. 9.

**Data availability.** Coordinates and structure factors of RNF168 UDM1–K63-Ub$_2$ (form I), RNF168 UDM1–K63-Ub$_2$ (form II), RNF168 UDM1–K63-Ub$_2$ (tetrameric form), and RNF168 UDM2ΔC–K63-Ub$_2$ have been deposited in the Protein Data Bank under accession codes 5XIS, 5XIT, 5YDK, and 5XIU, respectively. Other data are available from the corresponding authors upon reasonable request.

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

## Acknowledgements

We thank Drs. Takatoshi Arakawa and Shinya Fushinobu for technical help during CD measurement. We thank the beamline staff of macromolecular crystallography beamlines of Photon Factory (Tsukuba, Japan) and BL41XU of SPring-8 (Hyogo, Japan) for technical help during data collection. We also thank CentMeRE, Graduate School of Medicine, Osaka University for technical assistance. This work was supported by Grant-in-Aid for Scientific Research on Innovative Areas 22121003 (S.F.), 15H01175 (Y.S.), 15H01183 (S.N.), Grant-in-Aid for Scientific Research (A) 24247014 (S.F.) and 26241014 (S.N.), Grant-in-Aid for Young Scientists (A) 24687012 (Y.S.), Grant-in-Aid for Scientific Research (B) 16H04750 (Y.S.), Grant-in-Aid for Scientific Research (C) T17K00550 (Y. H.), JST CREST JPMJCR1XMX5 (S.F.), Grant-in-Aid for JSPS fellows 15F15386 (T.T.) and The Daiichi Mitsubishi Foundation (S.N.).

## Author contributions

T.T. performed the crystallography and interaction analyses and wrote the paper. A. Tomita., Y.H. and S.N. established the cell lines and performed the cell biological experiments. Y.S., A.Toma, A.Y. and S.G.-I. assisted with the crystallography and interaction analyses. Y.S. edited the paper. All authors discussed the results. Y.S., S.F. and S.N. designed the experiments. S.F. and S.N. supervised the work and wrote the paper.

## Additional information

**Competing interests:** The authors declare no competing financial interests.

