## [Peer Review File · Nature Communications]

Reviewers' comments:

Reviewer #1 (Remarks to the Author):

The manuscript by Takahashi et al. describes structural studies of two modules identified in E3 ubiquitin ligase RNF168, UDM1 and UDM2, which specifically recognise K63-linked polyubiquitin. By first performing biochemical and biophysical characterisation of UDM1 and UDM2, the authors define the critical elements within these modules that support interactions with Lys63-linked ubiquitin chains. The authors further present crystal structures of the UDM1: K63-Ub2 and truncated UDM2: K63-Ub2 complexes, and gain molecular insights into recognition of K63 ubiquitin chains. They address functional relevance of K63 polyubiquitin binding by measuring the ability of RNF168 variants to form IR induced foci.

The manuscript presents some very interesting data and provides new insights into the modes that specific polyubiquitin binding domains/motifs employ to ensure linkage specificity. It also furthers our understanding of RNF168 function and suggests the mechanism by which RNF168 may recognise polyubiquitylated histone H1. Overall, the data are of interest to the community and the wider field. However, given the previous studies on RNF168, there is a slight expectation that the manuscript should go a bit further, and could be improved by some restructuring and a few additional experiments (please see comments below). Furthermore, it would benefit from inclusion of the additional background information that would provide a meaningful context for the structural data and facilitate their interpretation. Importantly, it should also address the key questions about how UDM1 and UDM2 achieve K63 linkage specificity with more clarity.

Specific comments are listed below:

-Abstract is very difficult to read and the relationships between different motifs and modules are hard to grasp, especially since many different acronyms are used (UDM1, UDM2, LRM1, UMI, UAD and MIU2).

-Page 5: "Previous studies have shown that RNF168 UDM1 comprises UMI (residues141–156) and MIU1 (residues 171–188) as Ub-binding motifs."
Please provide references.

-Supplementary Fig. 1a
It is not clear what the images represent (Coomassie-stained gels? Western blots?).

-Page 5: "However, the obtained complex structure could not be interpreted as a physiologically relevant form."
Could the authors comment on why this was the case?

-The authors describe how LRM1-UMI and UAD-MIU2 interact with K63-Ub2, but molecular basis of specificity for K63-Ub2 is not clearly outlined. How can poor affinities for M1-Ub2 and K48-Ub2 be explained by the structural data? Equally, what would preclude binding of Ub2 of other linkages? These questions are only partially addressed in the paragraph entitled "Mechanisms for the linkage specificity of RNF168 UDM1 and UDM2". In addition, a non-specialist reader might find it difficult to understand how the distances to which the authors refer translate into linkage specificity.

-Not much information is provided on other ubiquitin binding domains and the mechanisms that they employ to ensure specificity.

-Page 9: "The C α – N ϵ distance between Leu71 of Ub dist and Lys63 of Ub prox is 13.2 Å...". The distance does not match Fig 4a (shown as 13.8 Å).

-Page 10, Fig. 5. The authors do not demonstrate that the siRNA against RNF168 downregulates expression of endogenous RNF168. The Figure should include Western blots against RNF168 with extracts from U2OS cells transfected with siCTRL and siRNF168. Also, only a single siRNA against RNF168 was used.

-page 10, Fig 5. Inducible expression of the siRNA-resistant RNF168 in retrovirus-transduced cells has not been demonstrated. Western blots showing doxycycline-induced expression of the siRNA-resistant RNF168 should be included.

-Fig. 5a. Why does the doxycycline induced RNF168 in cells transfected with siRNF168 show formation of foci in the absence of IR (Fig 5a, right)? Shouldn't its expression look similar to the expression of endogenous RNF168 in siCTRL cells (Fig 5a, left)?

-Fig. 5. The authors demonstrate formation of foci following IR. However, these may not necessarily correspond to sites of DNA damage. To demonstrate that RNF168 accumulates at sites of DNA double strand breaks the authors should measure colocalisation with γ -H2AX.

-Ideally, functional relevance of UDM1 and UDM2 could be strengthened by the additional functional assays, such as the cell survival experiments. Since the authors already have all the relevant cell lines (RNF168 depleted cells complemented with the different RNF168 variants used in Fig 5), their sensitivity to IR could be tested to provide a link between ubiquitin binding function of RNF168 and cell survival after genotoxic stress.

-page 11, Fig. 11. How were small and large foci defined and how was consistency ensured between the samples?

-page 11: "Mutations that impair binding of RNF168 UDM1 to K63-Ub2 (L116A in LRM1 and D175A in UMI)...."

Mutation L116A is included in Table 1 and its effect on K63-Ub2 is reported; however, mutation D175A is not included- what evidence do authors have that it impairs K63-Ub2 binding?

Reviewer #2 (Remarks to the Author):

Review of the Manuscript entitled: "Structural insights into two distinct binding modules for Lys63-linked polyubiquitin chains in RNF168" by Takahashi et al.

In the manuscript the authors present the crystal structures of RNF168 UDM1 and UDM2 domains bound to Lys63-linked di-Ubiquitin (K63-Ub2). The structures allow to determine the regions in UDM1 and UDM2 domains that interact specifically with either the distal or proximal Ub. The specificity of binding was verified using SPR measurements and the function of the UDM1 and UDM2 domains was tested in cells using immunofluorescence microscopy analysis

The authors did an extensive structural analysis of the K63-Ub2 bound to the UDM domains, which provides a sensible explanation of RNF168 specific recognition of Lys63-linkage. However the SPR and cell biological experiments raise some issues that need to be addressed before publication of this manuscript could be considered.

1. Some of the SPR sensograms in Sup. Fig. 1b show considerable signal decrease in response to analyte binding, for example the sensorgram with LRM1 as ligand and K63-Ub2 as analyte. That could mean that the analyte is binding more to the control flow cell than to the flow cell with the ligand. Normally small soluble proteins such as Ub2, if folded correctly, should not show strong, unspecific binding to the control flow cell under the described experimental condition. As all the analytes show this type of responses, their quality should be tested with respect to proper folding.

2. Based on the sensorgrams in Sup Fig 1b all the Ub2 have very fast dissociation rates that can't be properly estimated using Biacore T100. However, the authors calculated the equilibrium dissociation constants (K_d) using the 1:1 interaction model (M&M, page 27) that is estimating dissociation rate as part of the fitting process. In case of such a fast dissociation rate a steady state affinity model should be used instead of the kinetic fitting. The authors should clarify which part of the data was used for fitting and possibly provide the equation used.

3. As mentioned above, the Ub2, including K63-Ub2, have very fast dissociation rates in the SPR experiments. However, mixtures of K63-Ub2 with UDM1 and UDM2, prior to crystallization, were passed through a 16/60 Superdex75 column (M&M, page 25) to remove unbound K63-Ub2. Assuming that the dissociation rate, measured in the SPR experiment, are applicable also in solution, the complex between K63-Ub2 and UDM's should fall apart during the column separation. Due to the fast dissociation rate the molecules would have less chance to rebind and thus should be separated by their size. However K63-Ub2 and UDM's were crystallized in a complex, so obviously formed a stable complex during the size exclusion chromatography. It seems the complex probably has a slower dissociation rate in solution. This raises the question if the K_d calculated in the SPR experiments accurately reflect the affinities of the investigated complexes.

4. It is not clear to me why the Ub2 binding to the full UDM1 domain was not tested in SPR as it was done for UDM2 domain.

5. In Fig. 5a, judging from the representative cells shown as an example, the expression level of RNF168 in the siRNF168+RNF168*WT experiment without IR seems to me much higher than expression level of RNF168 in the siCTRL experiment without IR. However according to authors "Cells expressing high level of RNF168 were excluded from analysis" (M&M, page 28). The same is true for Sup. Fig 4. The expression level of RNF168 in D446A, S142 E443R, S142 R439A and S142A D446 cells seems to me higher than in in siCTRL control cell.

6. It is not clear to me how the authors classified the cell according to the foci size (M&M, page 28). How was the upper limit for the "small" foci defined?

7. The bar graph representation Fig. 5b and Sup. Fig. 4 is not the best way to present the immunofluorescence microscopy data since it does not give information about distributions. Dot plots of individual cells with number of foci on the Y-axis will be much more informative. Also, the authors should provide the information how the p-values, presented on page 11, were calculated.

8. The Western blot analysis of the endogenous RNF168 silencing and of the exogenous RNF168 expression (M&M, page 28) should be presented in supplemental materials.

9. I would advise the authors to give more details regarding the statement on page 5: "However, the obtained complex structure could not be interpreted as a physiologically relevant form"

10. Fig. 2b shows two different views on the region of contact, the transition/rotation between the two views is not very obvious. I would suggest to expand the visible fragments of the structure, so the

exact same residues will be present in both pictures and show degree of rotation between the pictures. Same holds true for Fig. 2c, Fig. 3b and Fig. 3d

11. Sup. Fig 2c. "Crystal packing of one of the two complexes from the asymmetric unit of the form I crystal" according to my understanding shows two complexes from two different asymmetric units. To avoid the confusing I would suggest to show two or more asymmetric units so the relative orientation of the molecules in the asymmetric unit and between them will be clearer. I would also suggest to do that for crystal form II

12. In the first paragraph of the section "Mechanisms for the linkage specificity of RNF168 UDM1 and UDM2", as I understood, the authors explained why only K63-Ub2 could bind the LRM1-UMI in the conformation observed in the crystal structures. However, I find that the paragraph would benefit from rephrasing to make this point more obvious to the reader.

13. The authors concentrate in the discussion on the model of RNF168 interaction with histones. However these interactions were not part of the results section.

Reviewer #3 (Remarks to the Author):

The manuscript " structural insights into two distinct binding modules for Lys63-linked polyubiquitin chains in RNF168" by Takahashi et al. reports the crystal structures of RNF168 UDM 1 & 2 bound to Lys63 Di-Ub together with biochemistry and cell biology to support the structure's conclusions. They suggest that UDM 1 and 2 are specific for Lys63-Di-UB and then explain the structural basis for this specificity. This work therefore advances our understanding of how RNF168 binds ubiquitin chains. However, the data supporting the conclusions require further validation.

All the binding experiments were done using a single method (SPR) with GST fusion proteins. GST forms dimer therefore it can introduce an artifact to the binding results (see: Sims et al. NSMB 16 883-889 (2009)). The authors should perform the binding experiment without GST, either with SPR or using a different method.

It is not clear why the authors did not test binding of Lys63 di-UB to UDM1 possessing not only LRM1 and UM1 but also MIU1. Also, they should test the binding with Lys63 linked tri-ub to see whether it affects binding. We cannot rule out the possibility that in the context of Tri ubiquitin chains all 3 domains are involved in the binding.

While structure based mutations do show some defects in binding, it is possible that these mutations are structural mutations that affect protein stability/folding. Authors should demonstrate, using CD or any other method, that these are not structural mutations.

Does the Lys63 di-Ub undergo conformational changes upon binding? Please provide superposition with free di-Ub structure.

Regarding the structure of UMI-MIU1 it is not clear what it means that the structure could not be interpreted as a physiologically relevant form. Please explain.

Since the Lys63 linkage is not seen in the structure, the authors have to show not only the ASU but also molecules outside the ASU to demonstrate that these are the right distal and proximal Ubiquitins that are linked together.

How can the occupancy of the linkage be 50%? All the molecules are Di-Ub, therefore the linkage is there. It is possible that it is flexible and not seen in the structure, but the occupancy should be 100%. Please explain this point.

Please design structure-based mutations on the Di-Ub and then demonstrate defects in binding.

This work lacks binding to K27 di ubiquitin. It has been shown that RNF168 promotes K27 chains thereby authors should also test the specificity of these chains to RNF168 UDM 1&2

The authors suggest that in the case of RNF169 the lack of Lys63 Di-Ub specificity is due to the

difference in the linker between LRM1 and UMI. This linker is shorter in? 5 AA compared to RNF168. Please show that swapping that linker with the one of RNF168 affects selectivity. The distances in fig. 4 do not fit the ones in the text.

Reviewer #4 (Remarks to the Author):

In their manuscript "Structural insights into two distinct binding modules for Lys63-linked polyubiquitin chains in RNF168" Takahashi and co-workers resolved the crystal structures of two Ubiquitin binding modules in complex with K63-di Ubiquitin consisting of three Ubiquitin binding domains of RNF168 and characterized the binding specificity with biochemical methods. They work could explain the preferential binding of RNF168 to K63 Polyubiquitin chains, for one domain LRM1 - UMI the authors propose a novel mechanism of chain-specific Ubiquitin binding, that is based on interaction with the distal Ubiquitin. In cellular assays the effect of specific point mutations on the recruitment of RNF168 to double strand breaks is examined. Here, only a mutation in UDM2 could diminish the recruitment of UDM2, the importance of UDM2 for RNF168 recruitment has been published before based on an other point mutation in very close distance. The study is in general well examined and the manuscript well written. The proposal of a new way of chain specific Ub binding is of great interest for researchers who are interested in Ubiquitin binding. Unfortunately, the study provides basically no new insights into the cellular activity of RNF168. As also stated by the authors, it is already known, that RNF168 bind preferentially K63 chains and that the UDM2 domain is essential for its recruitment. The authors could not show a specific function of UMI and UDM1. This study is better suited a journal specialized on structural biology.

Re: NCOMMS-17-11625-T

Comments from Reviewer #1:

-Abstract is very difficult to read and the relationships between different motifs and modules are hard to grasp, especially since many different acronyms are used (UDM1, UDM2, LRM1, UMI, UAD and MIU2).

We rewrote the abstract and removed most of the confusing acronyms.

-Page 5: "Previous studies have shown that RNF168 UDM1 comprises UMI (residues 141–156) and MIU1 (residues 171–188) as Ub-binding motifs." Please provide references.

We added the following references:

Pinato *et al.*, *Mol Cell Biol* 31, 118-26 (2011).

Doil *et al.*, *Cell* 136, 435-46 (2009).

Pinato *et al.*, *BMC Mol Biol* 10, 55 (2009).

Stewart *et al.*, *Cell* 136, 420-34 (2009).

-Supplementary Fig. 1a It is not clear what the images represent (Coomassie-stained gels? Western blots?).

We mentioned that the bound proteins were analyzed by SDS-PAGE with Coomassie brilliant blue staining in figure legends for Supplementary Fig. 1a.

-Page 5: "However, the obtained complex structure could not be interpreted as a physiologically relevant form." Could the authors comment on why this was the case?

In Supplementary Fig 2a, we showed the structure of the 2:2 tetrameric UDM1–K63-Ub₂ complex, where UMI and MIU1 interact with the distal and proximal Ub moieties in two different K63-Ub₂. However, this tetrameric assembly in the crystal is inconsistent with the molar mass determined by size-exclusion chromatography coupled with multi-angle laser light scattering. Therefore, we concluded that the tetrameric assembly of UDM1–K63-Ub₂ in the crystal is an artifact. These points are described in the subsection "Structure of RNF168 UDM1 in complex with K63-Ub₂" in pg 6.

-The authors describe how LRM1-UMI and UAD-MIU2 interact with K63-Ub2, but molecular basis of specificity for K63-Ub2 is not clearly outlined. How can poor affinities for M1-Ub2 and K48-Ub2 be explained by the structural data? Equally, what would preclude binding of Ub2 of other linkages? These questions are only partially addressed in the paragraph entitled “Mechanisms for the linkage specificity of RNF168 UDM1 and UDM2”. In addition, a non-specialist reader might find it difficult to understand how the distances to which the authors refer translate into linkage specificity.

We rewrote the subsection “Mechanisms for the linkage specificity of RNF168 UDM1 and UDM2” to improve clarity. In the beginning of this subsection, we added the explanation as to how the structural mechanism of the linkage specificity has been discussed as follows:

“Previous structural studies on linkage-specific UBDs have shown that the relative spacing and orientations of their Ub^{dist}- and Ub^{prox}-interacting surfaces determine the specificities to certain linkage types of Ub chains: the bound Ub^{dist} and Ub^{prox} are fixed on the linkage-specific UBDs and thereby only specific lysine residue(s) and/or Met1 of Ub^{prox} can be physically connected to Gly76 of Ub^{dist}. Since the C-terminal tail of Ub^{dist} (residues 71–76) is flexible, the structural mechanism of the linkage specificity of UBDs has been discussed on the basis of the length between Leu71 of Ub^{dist} (the first residue of the C-terminal tail) and lysine residues or Met1 of Ub^{prox} in the UBD-bound diUb structure.”

-Not much information is provided on other ubiquitin binding domains and the mechanisms that they employ to ensure specificity.

We added information about other UBDs in Discussion.

-Page 9: “The C α – N ϵ distance between Leu71 of Ub dist and Lys63 of Ub prox is 13. 2 Å...”. The distance does not match Fig 4a (shown as 13.8 Å).

We corrected this, accordingly.

-Page 10, Fig. 5. The authors do not demonstrate that the siRNA against RNF168 downregulates expression of endogenous RNF168. The Figure should include Western blots against RNF168 with

extracts from U2OS cells transfected with siCTRL and siRNF168. Also, only a single siRNA against RNF168 was used.

We carried out the corresponding Western blots and added experiments using another siRNA against RNF168 (Supplementary Fig. 6)

-page 10, Fig 5. Inducible expression of the siRNA-resistant RNF168 in retrovirus-transduced cells has not been demonstrated. Western blots showing doxycycline-induced expression of the siRNA-resistant RNF168 should be included.

We carried out the corresponding Western blots (Supplementary Fig. 6) and confirmed the doxycycline-induced expression of RNF168 proteins. Furthermore, in the IF experiment, we carefully inspected the expression of RNF168 proteins in the individual cells and analyzed the foci formation in the only cells that obviously expressed RNF168. The result of our IF experiment should be convincing.

-Fig. 5a. Why does the doxycycline induced RNF168 in cells transfected with siRNF168 show formation of foci in the absence of IR (Fig 5a, right)? Shouldn't its expression look similar to the expression of endogenous RNF168 in siCTRL cells (Fig 5a, left)?

In G1 cells, small numbers of DSBs spontaneously occur during cell division, even without extrinsic factors causing DNA lesions (Lukas et al. Nat Cell Biol. ;13:243-53. doi: 10.1038/ncb2201.2011) . RNF168 can be accumulated to these naturally occurring DSBs. Therefore, small numbers of foci can be observed in cells without IR, as shown in Supplementary Fig. 6f, g (tiny dots that look like foci are backgrounds).

For counting the number of foci, we carefully selected cells expressing doxycycline-induced RNF168 in a level similar to endogenous RNF168, and excluded cells excessively expressing RNF168. The expression level of doxycycline-induced RNF168 could be successfully controlled by optimizing the doxycycline concentration, and was adjusted to the level similar to the endogenous RNF168 expression. Concomitantly, we needed to increase the sensitivity of the confocal microscope to detect foci. In this high sensitivity condition, the signal range became narrower and thereby a small difference in the expression level can be exaggerated. We therefore realize that the expression level of doxycycline-induced RNF168 is actually close to that of endogenous RNF168.

-Fig. 5. The authors demonstrate formation of foci following IR. However, these may not necessarily correspond to sites of DNA damage. To demonstrate that RNF168 accumulates at sites of DNA double strand breaks the authors should measure colocalisation with γ -H2AX. -Ideally, functional relevance of UDM1 and UDM2 could be strengthened by the additional functional assays, such as the cell survival experiments. Since the authors already have all the relevant cell lines (RNF168 depleted cells complemented with the different RNF168 variants used in Fig 5), their sensitivity to IR could be tested to provide a link between ubiquitin binding function of RNF168 and cell survival after genotoxic stress.

Our new experiments showed that almost all RNF168 foci and RNF168 mutant foci clearly co-localized with γ -H2AX foci (Fig. 5 and Supplementary Fig. 6). On the other hand, the suggested cell survival experiments were technically impossible. As the number of the cells that express doxycycline-induced siRNA-resistant RNF168 gradually decrease during culture, even without IR, we could not assess the functional relevance of UDM1 or UDM2 to cell survival after IR.

-page 11, Fig. 11. How were small and large foci defined and how was consistency ensured between the samples?

We carried out new experiments, where we did not distinguish between small and large foci. We counted the number of foci in a single cell. Therefore, we removed descriptions regarding the size of foci.

-page 11: "Mutations that impair binding of RNF168 UDM1 to K63-Ub2 (L116A in LRMI and D175A in UMI)...." Mutation L116A is included in Table 1 and its effect on K63-Ub2 is reported; however, mutation D175A is not included- what evidence do authors have that it impairs K63-Ub2 binding?

We examined the interaction between the D175A mutant and diUb species (K63-, M1- and K48-Ub₂) by SPR analysis (Table 1 and Supplementary Fig. 1).

Comments from Reviewer #2 (Remarks to the Author):

... The authors did an extensive structural analysis of the K63-Ub2 bound to the UDM domains, which provides a sensible explanation of RNF168 specific recognition of Lys63-linkage. However the SPR and cell biological experiments raise some issues that need to be addressed before publication of this manuscript could be considered.

1. Some of the SPR sensograms in Sup. Fig. 1b show considerable signal decrease in response to analyte binding, for example the sensorgram with LRM1 as ligand and K63-Ub2 as analyte. That could mean that the analyte is binding more to the control flow cell than to the flow cell with the ligand. Normally small soluble proteins such as Ub2, if folded correctly, should not show strong, unspecific binding to the control flow cell under the described experimental condition. As all the analytes show this type of responses, their quality should be tested with respect to proper folding.

Met1-, Lys48- and Lys63-linked diubiquitin molecules are highly stable. Unfolding of these molecules need strong denaturant or very high temperature condition. Also, the diubiquitin molecules prepared in this study have been used for other crystallization experiments and successfully produced crystals with or without Ub-binding proteins. During crystallization experiments, drops are usually clear before starting crystallization, suggesting the diubiquitin molecules we prepared are of high quality. Altogether, we are confident that our diubiquitin molecules are properly folded and suitable for SPR analysis.

As for SPR analysis, we have analyzed different UBDs and some of them showed signal decreases like LRM1. It is likely that the negative signal reflects some weak and non-specific binding between diubiquitin and GST. However, this does not affect the determination of K_d , because SPR analyses using GST-tagged protein and untagged protein showed similar results (Table 1 and Supplementary Fig. 1c).

2. Based on the sensorgrams in Sup Fig 1b all the Ub2 have very fast dissociation rates that can't be properly estimated using Biacore T100. However, the authors calculated the equilibrium dissociation constants (K_d) using the 1:1 interaction model (M&M, page 27) that is estimating dissociation rate as part of the fitting process. In case of such a fast dissociation rate a steady state affinity model should be used instead of the kinetic fitting. The authors should clarify which part of the data was used for fitting and possibly provide the equation used.

Dissociation constants (K_d) were calculated using a steady state affinity model (but not using a kinetic model). We clearly mentioned this point in “Methods” in the revised manuscript.

3. As mentioned above, the Ub₂, including K63-Ub₂, have very fast dissociation rates in the SPR experiments. However, mixtures of K63-Ub₂ with UDM1 and UDM2, prior to crystallization, were passed through a 16/60 Superdex75 column (M&M, page 25) to remove unbound K63-Ub₂. Assuming that the dissociation rate, measured in the SPR experiment, are applicable also in solution, the complex between K63-Ub₂ and UDM's should fall apart during the column separation. Due to the fast dissociation rate the molecules would have less chance to rebind and thus should be separated by their size. However K63-Ub₂ and UDM's were crystallized in a complex, so obviously formed a stable complex during the size exclusion chromatography. It seems the complex probably has a slower dissociation rate in solution. This raises the question if the K_d calculated in the SPR experiments accurately reflect the affinities of the investigated complexes.

The SPR sensorgrams in this study show not only fast dissociation but also fast association between K63-Ub₂ and UDM1 or UDM2. In our experience, binding behavior in SPR analysis is not always related to behavior in size-exclusion chromatography. We agree that a protein complex showing fast association-fast dissociation in SPR analysis tends to fall apart during size-exclusion chromatography. However, practically, this is not always the case.

4. It is not clear to me why the Ub₂ binding to the full UDM1 domain was not tested in SPR as it was done for UDM2 domain.

We have performed SPR analysis with the full-length UDM1. However, since both the N-terminal LRM1-UMI and C-terminal MIU1 could bind Ub₂, the observed spectrograms could not fit in a 1:1 interaction model. The equation $R_{eq} = Concentration * R_{max} / (K_d + Concentration) + Offset$, only applies to 1:1 interaction model, and could not fit our binding data from a 2:1 interaction. Instead, we added the dissociation constant of MIU1, to evaluate how MIU1 and LRM1-UMI contribute to Ub-binding.

5. In Fig. 5a, judging from the representative cells shown as an example, the expression level of RNF168 in the siRNF168+RNF168*WT experiment without IR seems to me much higher than expression level of RNF168 in the siCTRL experiment without IR. However according to authors "Cells expressing high level of RNF168 were excluded from analysis" (M&M, page 28). The same is true for Sup. Fig 4. The expression level of RNF168 in D446A, S142 E443R, S142 R439A and S142A D446 cells seems to me higher than in siCTRL control cell.

We carried out Western blotting to measure the protein level of RNF168 in U2OS cells (Supplementary Fig. 6).

In G1 cells, small numbers of DSBs spontaneously occur during cell division, even without extrinsic factors causing DNA lesions (Lukas et al. Nat Cell Biol. ;13:243-53. doi: 10.1038/ncb2201.2011) . RNF168 can be accumulated to these naturally occurring DSBs. Therefore, small numbers of foci can be observed in cells without IR, as shown in Supplementary Fig. 6f, g (tiny dots that look like foci are backgrounds).

For counting the number of foci, we carefully selected cells expressing doxycycline-induced RNF168 in a level similar to endogenous RNF168, and excluded cells excessively expressing RNF168. The expression level of doxycycline-induced RNF168 could be successfully controlled by optimizing the doxycycline concentration, and was adjusted to the level similar to the endogenous RNF168 expression. Concomitantly, we needed to increase the sensitivity of the confocal microscope to detect foci. In this high sensitivity condition, the signal range became narrower and thereby a small difference in the expression level can be exaggerated. We therefore realize that the expression level of doxycycline-induced RNF168 is actually close to that of endogenous RNF168.

6. It is not clear to me how the authors classified the cell according to the foci size (M&M, page 28). How was the upper limit for the “small” foci defined?

We carried out new experiments, where we did not distinguish between small and large foci. We counted the number of foci in a single cell. Therefore, we removed descriptions regarding the size of foci.

7. The bar graph representation Fig. 5b and Sup. Fig. 4 is not the best way to present the immunofluorescence microscopy data since it does not give information about distributions. Dot plots of individual cells with number of foci on the Y-axis will be much more informative. Also, the authors should provide the information how the p-values, presented on page 11, were calculated.

We presented the corresponding dot plots in Fig. 5 and Supplementary Fig. 6e, and described the calculation method for *p*-values in “Methods” and figure legends.

8. The Western blot analysis of the endogenous RNF168 silencing and of the exogenous RNF168 expression (M&M, page 28) should be presented in supplemental materials.

The corresponding Western blot analysis is shown in Supplementary Fig. 6.

9. I would advise the authors to give more details regarding the statement on page 5: “However, the obtained complex structure could not be interpreted as a physiologically relevant form”

In Supplementary Fig 2a, we showed the structure of the 2:2 tetrameric UDM1–K63-Ub₂ complex, where UMI and MIU1 interact with the distal and proximal Ub moieties in two different K63-Ub₂. However, this tetrameric assembly in the crystal is inconsistent with the molar mass determined by size-exclusion chromatography coupled with multi-angle laser light scattering. Therefore, we concluded that the tetrameric assembly of UDM1–K63-Ub₂ in the crystal is an artifact. These points are described in the subsection “Structure of RNF168 UDM1 in complex with K63-Ub₂” in pg 6.

10. Fig. 2b shows two different views on the region of contact, the transition/rotation between the two views is not very obvious. I would suggest to expand the visible fragments of the structure, so the exact same residues will be present in both pictures and show degree of rotation between the pictures. Same holds true for Fig. 2c, Fig. 3b and Fig. 3d

We revised the corresponding figures, accordingly (Figs. 2b,c and 3b,d in the revised manuscripts).

11. Sup. Fig 2c. “Crystal packing of one of the two complexes from the asymmetric unit of the form I crystal” according to my understanding shows two complexes from two different asymmetric units. To avoid the confusing I would suggest to show two or more asymmetric units so the relative orientation of the molecules in the asymmetric unit and between them will be clearer. I would also suggest to do that for crystal form II

We showed the two asymmetric units in form I and form II crystals in Supplementary Fig. 2b.

12. In the first paragraph of the section “Mechanisms for the linkage specificity of RNF168 UDM1 and UDM2”, as I understood, the authors explained why only K63-Ub₂ could bind the LRMI-UMI in

the conformation observed in the crystal structures. However, I find that the paragraph would benefit from rephrasing to make this point more obvious to the reader.

We rewrote the subsection “Mechanisms for the linkage specificity of RNF168 UDM1 and UDM2” to improve clarity. In the beginning of this subsection, we added the explanation as to how the structural mechanism of the linkage specificity has been discussed as follows:

“Previous structural studies on linkage-specific UBDs have shown that the relative spacing and orientations of their Ub^{dist}- and Ub^{prox}-interacting surfaces determine the specificities to certain linkage types of Ub chains: the bound Ub^{dist} and Ub^{prox} are fixed on the linkage-specific UBDs and thereby only specific lysine residue(s) and/or Met1 of Ub^{prox} can be physically connected to Gly76 of Ub^{dist}. Since the C-terminal tail of Ub^{dist} (residues 71–76) is flexible, the structural mechanism of the linkage specificity of UBDs has been discussed on the basis of the length between Leu71 of Ub^{dist} (the first residue of the C-terminal tail) and lysine residues or Met1 of Ub^{prox} in the UBD-bound diUb structure.”

13. The authors concentrate in the discussion on the model of RNF168 interaction with histones. However these interactions were not part of the results section.

We toned down the discussion about the interaction between RNF168 and histones, and removed the corresponding schematic figures, accordingly.

Reviewer #3 (Remarks to the Author):

...This work therefore advances our understanding of how RNF168 binds ubiquitin chains. However, the data supporting the conclusions require further validation.

All the binding experiments were done using a single method (SPR) with GST fusion proteins. GST forms dimer therefore it can introduce an artifact to the binding results (see: Sims et al. NSMB 16 883-889 (2009)). The authors should perform the binding experiment without GST, either with SPR or using a different method.

We carried out the pull-down experiments using His₆-tagged proteins to prevent the dimer formation by GST tags (Supplementary Fig. 1a). On the other hand, in the SPR analysis, we confirmed that the GST tag did not affect the determination of K_d by comparing the untagged UBDs (LRM1-UMI and UAD-MIU2) with the GST-tagged UBDs. As shown in Table 1, the GST-tagged

LRM1-UMI and UAD-MIU2 show linkage specificities similar to the untagged LRM1-UMI and UAD-MIU2.

It is not clear why the authors did not test binding of Lys63 di-UB to UDMI possessing not only LRM1 and UMI but also MIU1. Also, they should test the binding with Lys63 linked tri-ub to see whether it affects binding. We cannot rule out the possibility that in the context of Tri ubiquitin chains all 3 domains are involved in the binding.

We have performed SPR analysis with the full-length UDMI. However, since both the N-terminal LRM1-UMI and C-terminal MIU1 could bind Ub₂, the observed spectrograms could not fit in a 1:1 interaction model. The equation $R_{eq} = Concentration * R_{max} / (K_d + Concentration) + Offset$, only applies to 1:1 interaction model, and could not fit our binding data from a 2:1 interaction. Instead, we added the dissociation constant of MIU1, to evaluate how MIU1 and LRM1-UMI contribute to Ub-binding.

While structure based mutations do show some defects in binding, it is possible that these mutations are structural mutations that affect protein stability/folding. Authors should demonstrate, using CD or any other method, that these are not structural mutations.

We carried out the corresponding CD experiments and confirmed that all mutants are folded similarly. The data are shown in Supplementary Fig. 8.

Does the Lys63 di-Ub undergo conformational changes upon binding? Please provide superposition with free di-Ub structure.

We showed the crystal structure of the apo-form K63-Ub₂ for comparison in Supplementary Fig. 5.

Regarding the structure of UMI-MIU1 it is not clear what it means that the structure could not be interpreted as a physiologically relevant form. Please explain.

In Supplementary Fig 2a, we showed the structure of the 2:2 tetrameric UDMI-K63-Ub₂ complex, where UMI and MIU1 interact with the distal and proximal Ub moieties in two different

K63-Ub₂. However, this tetrameric assembly in the crystal is inconsistent with the molar mass determined by size-exclusion chromatography coupled with multi-angle laser light scattering. Therefore, we concluded that the tetrameric assembly of UDM1–K63-Ub₂ in the crystal is an artifact. These points are described in the subsection “Structure of RNF168 UDM1 in complex with K63-Ub₂” in pg 6.

Since the Lys63 linkage is not seen in the structure, the authors have to show not only the ASU but also molecules outside the ASU to demonstrate that these are the right distal and proximal Ubiquitins that are linked together. How can the occupancy of the linkage be 50%? All the molecules are Di-Ub, therefore the linkage is there. It is possible that it is flexible and not seen in the structure, but the occupancy should be 100%. Please explain this point.

The asymmetric unit contains one ubiquitin, indicating that both proximal and distal ubiquitin moieties of K63-Ub₂ display the same structure in the crystal. As a consequence, the electron density of ubiquitin is the average of the proximal and distal ubiquitin moieties. The C-terminal tail of the distal Ub is linked to Lys63 of the proximal ubiquitin, but the C-terminal tail of the distal Ub is not. This may explain why the C-terminal tail of Ub is not visible in the crystal structure. This point is explained in the subsection “Structure of RNF168 UDM2ΔC in complex with K63-Ub₂” in pg. 9.

Please design structure-based mutations on the Di-Ub and then demonstrate defects in binding.

Mutations of K63-Ub₂ are technically difficult. Although Ub chains can be generated by multiple enzymatic reactions, mutations of Ub may inhibit these reactions. In this study, we focus on the identification and characterization of two K63-linkage-specific Ub-binding motifs, and believe that the current data are sufficient for this purpose. Therefore, we are willing to omit this suggested experiment.

This work lacks binding to K27 di ubiquitin. It has been shown that RNF168 promotes K27 chains thereby authors should also test the specificity of these chains to RNF168 UDM 1&2

K27-Ub₂ is much less stable than other diubiquitin species and tends to aggregate, affecting the result of the binding assay. Therefore, we used lower amount (1.5 μg) of K27-Ub₂ for the binding assay. The result is shown in Supplementary Fig. 1b.

The authors suggest that in the case of RNF169 the lack of Lys63 Di-Ub specificity is due to the difference in the linker between LRM1 and UMI. This linker is shorter in? 5 AA compared to RNF168. Please show that swapping that linker with the one of RNF168 affects selectivity.

We replaced the LRM1-UMI linker of RNF168 by that of RNF169 and tested the binding to K48- or K63-Ub₂ by pull-down assay. The RNF168 mutant containing the RNF169 linker abolished the binding to K63-Ub₂. This result was shown in Supplementary Fig. 7c.

The distances in fig. 4 do not fit the ones in the text.

We corrected this discrepancy.

Reviewer #4 (Remarks to the Author):

... The study is in general well examined and the manuscript well written. The proposal of a new way of chain specific Ub binding is of great interest for researchers who are interested in Ubiquitin binding. Unfortunately, the study provides basically no new insights into the cellular activity of RNF168. As also stated by the authors, it is already known, that RNF168 bind preferentially K63 chains and that the UDM2 domain is essential for its recruitment. The authors could not show a specific function of UMI and UDMI. This study is better suited a journal specialized on structural biology.

In general, structural mechanisms are as important as cellular activity to completely understand biological processes. Therefore, we disagree with his/her opinion that structural studies are better suited for a journal specialized on structural biology.

REVIEWERS' COMMENTS:

Reviewer #1 (Remarks to the Author):

The authors have addressed majority of my concerns.
Minor point: Leu71 label is missing in Supplementary Fig. 5.

Reviewer #2 (Remarks to the Author):

The authors did extensive and thorough work to answer the comments. The extra experiments, additional data and corrections made the manuscript more clear and precise. In general I support publication of the manuscript in Nature Communications, however, a few minor issues, detailed below, should still be addressed before publication.

Authors answer to comment 1

Met1-, Lys48- and Lys63-linked diubiquitin molecules are highly stable. Unfolding of these molecules need strong denaturant or very high temperature condition. Also, the diubiquitin molecules prepared in this study have been used for other crystallization experiments and successfully produced crystals with or without Ub-binding proteins. During crystallization experiments, drops are usually clear before starting crystallization, suggesting the diubiquitin molecules we prepared are of high quality.

Altogether, we are confident that our diubiquitin molecules are properly folded and suitable for SPR analysis.

As for SPR analysis, we have analyzed different UBDs and some of them showed signal decreases like LRM1. It is likely that the negative signal reflects some weak and non-specific binding between diubiquitin and GST. However, this does not affect the determination of K_d , because SPR analyses using GST-tagged protein and untagged protein showed similar results (Table 1 and Supplementary Fig. 1c).

Comment

The authors didn't specify the procedure for the diubiquitin preparation. Depending on the preparation protocol some fraction of the diubiquitins could not be properly folded, and the relative quantity of this fraction can vary for different diubiquitin species. This fraction can give the unspecific signal during the SPR experiments. The ability of the diubiquitins to co-crystallise doesn't prove the purity or quality of the stock since only properly folded molecules would co-crystallise and the improperly folded could stay soluble in the solution.

Authors answer to comment 3

The SPR sensorgrams in this study show not only fast dissociation but also fast association between K63-Ub2 and UDM1 or UDM2. In our experience, binding behavior in SPR analysis is not always related to behavior in size-exclusion chromatography. We agree that a protein complex showing fast association-fast dissociation in SPR analysis tends to fall apart during size-exclusion chromatography. However, practically, this is not always the case

Comment

For the simple 1:1 interaction dissociation is dependent on dissociation rate constant (k_d) in contrast to association that is dependent on association (k_a) and dissociation rate constants. While k_d is

dependent on interaction k_a normally is only limited by the diffusion rate. In the experiments presented in the manuscript the k_d can't be estimated since it is too fast for detection rate of Biacore T200, so k_a can't be calculated either. The size-exclusion chromatography does not provide direct information regarding k_a or k_d . The only conclusion that can be made from it is that dissociation is slow enough to keep the complex together. So the question regarding whether the calculated K_d based on SPR experiments reflect the actual affinities of the investigated complexes still holds. The SPR results are in general well supported by the rest of the experiments and the overall conclusion holds. Nonetheless, the reported K_d values, if not confirmed by another type of binding assays, should rather be treated as apparent.

Authors answer to comment 4

We have performed SPR analysis with the full-length UDM1. However, since both the N-terminal LRM1-UMI and C-terminal MIU1 could bind Ub2, the observed spectrograms could not fit in a 1:1 interaction model. The equation $Req = \text{Concentration} * R_{max} / (K_d + \text{Concentration}) + \text{Offset}$, only applies to 1:1 interaction model, and could not fit our binding data from a 2:1 interaction. Instead, we added the dissociation constant of MIU1, to evaluate how MIU1 and LRM1-UMI contribute to Ub-binding.

Comment

If the authors suspect a 2:1 interaction where binding events are independent from each other, they can treat the equilibrium response Req as the results of the equilibrium of two independent binding events, and use following equation to fit the data:

$$Req = \text{Concentration} * R_{max} (\text{LMR1-UMI}) / (K_d (\text{LMR1-UMI}) + \text{Concentration}) + \text{Concentration} * R_{max} (\text{MIU1}) / (K_d (\text{MIU1}) + \text{Concentration}) + \text{Offset}$$

And then check if the calculated K_d 's agree with the K_d 's calculated for binding to LMR1-UMI and MIU1 separately.

Authors answer to comment 7

We presented the corresponding dot plots in Fig. 5 and Supplementary Fig. 6e, and described the calculation method for p-values in "Methods" and figure legends.

Comment

For statistical analysis authors used one-way ANOVA with post-hoc Tukey's HSD test. However due to the low limit problem the distributions of WT (no IR), D446A, S142A E433R, S142A D446A, L116A R439A and D175A R439A in Fig. 5b and WT (no IR), D446A, S142A E433R and S142A D446A in Supplementary Fig. 6e are definitely not normal. So I suggest to the authors to use a non-parametric method like, for example, Kruskal-Wallis test, to analyze the data. It probably won't change the outcome but will reduce the significance.

Authors answer to comment 9

In Supplementary Fig 2a, we showed the structure of the 2:2 tetrameric UDM1-K63-Ub2 complex, where UMI and MIU1 interact with the distal and proximal Ub moieties in two different K63-Ub2. However, this tetrameric assembly in the crystal is inconsistent with the molar mass determined by size-exclusion chromatography coupled with multi-angle laser light scattering. Therefore, we concluded that the tetrameric assembly of UDM1-K63-Ub2 in the crystal is an artifact. These points are described in the subsection "Structure of RNF168 UDM1 in complex with K63-Ub2" in pg 6.

Comment

The existence of 2:2 tetrameric form could be due of the high concentration of the proteins during crystallization. The concentration during size exclusion chromatography was much lower. I would suggest to the authors to analyze macromolecular interfaces in all there crystal forms using PISA (Proteins, Interfaces, Structures and Assemblies)

Reviewer #3 (Remarks to the Author):

The authors have satisfied my concerns

Re: NCOMMS-17-11625-A

Comments from Reviewer #1:

The authors have addressed majority of my concerns.

Minor point: Leu71 label is missing in Supplementary Fig. 5.

Leu71 label was added in Supplementary Fig. 5

Comments from Reviewer #2:

The authors didn't specify the procedure for the diubiquitin preparation. Depending on the preparation protocol some fraction of the diubiquitins could not be properly folded, and the relative quantity of this fraction can vary for different diubiquitin species. This fraction can give the unspecific signal during the SPR experiments. The ability of the diubiquitins to co-crystallise doesn't prove the purity or quality of the stock since only properly folded molecules would co-crystallise and the improperly folded could stay soluble in the solution.

We described the method for the diubiquitin preparation in the Methods section.

For the simple 1:1 interaction dissociation is dependent on dissociation rate constant (k_d) in contrast to association that is dependent on association (k_a) and dissociation rate constants. While k_d is dependent on interaction k_a normally is only limited by the diffusion rate. In the experiments presented in the manuscript the k_d can't be estimated since it is too fast for detection rate of Biacore T200, so k_a can't be calculated either. The size-exclusion chromatography does not provide direct information regarding k_a or k_d . The only conclusion that can be made from it is that dissociation is slow enough to keep the complex together. So the question regarding whether the calculated K_d based on SPR experiments reflect the actual affinities of the investigated complexes still holds.

The SPR results are in general well supported by the rest of the experiments and the overall conclusion holds. Nonetheless, the reported K_d values, if not confirmed by another type of binding assays, should rather be treated as apparent.

We basically agree with these comments based on the theoretical backgrounds of SPR analysis and size-exclusion chromatography. The K_d values determined in our study may be apparent. Difference in buffer conditions (*e.g.*, salt and detergent concentrations) between the SPR analysis and size-exclusion chromatography might affect the strength of the UDM1–Ub₂ and UDM2–Ub₂ binding.

On the other hand, the main purposes of the SPR analysis in our study are testing the linkage specificities of UBDs and the effects of structure-guided site-directed mutations on the binding between UBD and Ub₂ to assess the contribution of each interacting residue to the total binding. The K_d values determined in our study should be sufficient for these purposes.

If the authors suspect a 2:1 interaction where binding events are independent from each other, they can treat the equilibrium response Req as the results of the equilibrium of two independent binding events, and use following equation to fit the data:

$$Req = \text{Concentration} * Rmax (LMR1-UMI) / (Kd (LMR1-UMI) + \text{Concentration}) + \text{Concentration} * Rmax (MIU1) / (Kd (MIU1) + \text{Concentration}) + \text{Offset}$$

And then check if the calculated K_d 's agree with the K_d 's calculated for binding to LMR1-UMI and MIU1 separately.

We tested the suggested equation but failed to fit it well to the data with the K_d (LMR1–UMI) and K_d (MIU1) values determined separately. It seems that two binding events mediated by LMR1–UMI and MIU1 are not perfectly independent.

For statistical analysis authors used one-way ANOVA with post-hoc Tukey's HSD test. However due to the low limit problem the distributions of WT (no IR), D446A, S142A E433R, S142A D446A, L116A R439A and D175A R439A in Fig. 5b and WT (no IR), D446A,

S142A E433R and S142A D446A in Supplementary Fig. 6e are definitely not normal. So I suggest to the authors to use a non-parametric method like, for example, Kruskal–Wallis test, to analyze the data. It probably won't change the outcome but will reduce the significance.

We statistically analyzed the data by the Kruskal–Wallis test (Figure 5 and Supplementary Figure 6).

The existence of 2:2 tetrameric form could be due of the high concentration of the proteins during crystallization. The concentration during size exclusion chromatography was much lower. I would suggest to the authors to analyze macromolecular interfaces in all there crystal forms using PISA (Proteins, Interfaces, Structures and Assemblies)

We analyzed the buried surface areas by PISA. The calculated values are shown in the subsection “Structure of RNF168 UDM1 in complex with K63-Ub₂” (pg. 6).

Comment from Reviewer #3:

The authors have satisfied my concerns